# Heterostructure particles enable omnidispersible in water and oil towards organic dye recycle

Yongyang Song [1,2,7], Jiajia Zhou[3,7], Zhongpeng Zhu[4], Xiaoxia Li[1,2], Yue Zhang[1,2], Xinyi Shen[1,2], Padraic O'Reilly [5], Xiuling Li[6], Xinmiao Liang[6], Lei Jiang[1,2,4] & Shutao Wang [1,2,4] ✉

Dispersion of colloidal particles in water or oil is extensively desired for industrial and environmental applications. However, it often strongly depends on indispensable assistance of chemical surfactants or introduction of nano-protrusions onto the particle surface. Here we demonstrate the omnidispersity of hydrophilic-hydrophobic heterostructure particles (HL-HBPs), synthesized by a surface heterogeneous nanostructuring strategy. Photo-induced force microscopy (PiFM) and adhesion force images both indicate the heterogeneous distribution of hydrophilic domains and hydrophobic domains on the particle surface. These alternating domains allow HL-HBPs to be dispersed in various solvents with different polarity and boiling point. The HL-HBPs can efficiently adsorb organic dyes from water and release them into organic solvents within several seconds. The surface heterogeneous nanostructuring strategy provides an unconventional approach to achieve omnidispersion of colloidal particles beyond surface modification, and the omnidispersible HL-HBPs demonstrate superior capability for dye recycle merely by solvent exchange. These omnidispersible HL-HBPs show great potentials in industrial process and environmental protection.

Well-dispersed colloidal particles have been long pursued in extensive fields, they act as building blocks for self-assembly[1–3], as catalysts for energy conversion[4,5], as inks for visualization[6], and as imaging, diagnostic, and therapeutic agents for tumor management[7]. However, it remains a challenge to disperse particles into incompatible liquids[8,9], for example, dispersing carbon-based nanoparticles into polymer melts for photovoltaic device fabrication[10], dispersing hydrophobic photoinitiator nanoparticles in water for hydrogel printing[11], dispersing hydrophobic adsorbents in water for oil removal[12], and dispersing metal/inorganic nanoparticles in salt melts for energy storage and

transfer[13]. Surface modification is a classical strategy to balance the incompatibility between solids and liquids[14] (Fig. 1a). Chemical surfactants, including monomeric molecules[15], polymeric molecules[16–19], and biomacromolecules[20,21], have been pervasively utilized onto particle surface for well dispersion. They could enhance solid-liquid affinity, increase interparticle repulsive electrostatic interaction or steric hindrance, therefore allowing well dispersion of particles in desired liquids. An alternative approach seeks to modify the particle surface with nanoprotrusions, such as hedgehog-like particles[12,22] and raspberry-like particles[6]. The elaborately made nanoprotrusions could

[1]CAS Key Laboratory of Bio-inspired Materials and Interfacial Science, Technical Institute of Physics and Chemistry, Chinese Academy of Sciences, Beijing, P. R. China. [2]University of Chinese Academy of Sciences, Beijing, P. R. China. [3]South China Advanced Institute for Soft Matter Science and Technology, School of Emergent Soft Matter, South China University of Technology, Guangzhou, P. R. China. [4]Suzhou Institute for Advanced Research, University of Science and Technology of China, Suzhou, P. R. China. [5]Molecular Vista Inc., San Jose, CA, USA. [6]CAS Key Laboratory of Separation Science for Analytical Chemistry, Dalian Institute of Chemical Physics, Chinese Academy of Sciences, Dalian, P. R. China. [7]These authors contributed equally: Yongyang Song, Jiajia Zhou. ✉e-mail: stwang@mail.ipc.ac.cn

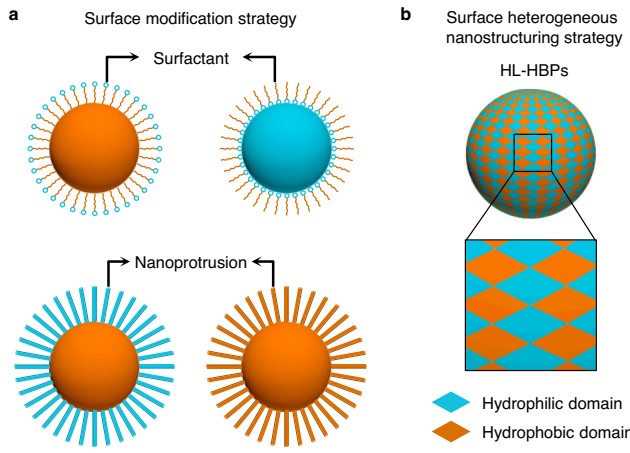

**a** Surface modification strategy

Surfactant

Nanoprotrusion

**b** Surface heterogeneous nanostructuring strategy

HL-HBPs

Hydrophilic domain
Hydrophobic domain

**Fig. 1 | Surface heterogeneous nanostructuring strategy for the design of omnidispersible hydrophilic-hydrophobic heterostructure particles (HL-HBPs). a** Conventional surface modification strategy for particle dispersion. Modification of surfactants or nanoprotrusions on particle surface is always utilized to obtain well-dispersed particles. **b** Surface heterogeneous nanostructuring strategy is proposed to enable particles' omnidispersity. For example, HL-HBPs exhibiting alternating hydrophilic domains and hydrophobic domains can realize well dispersity in both water and oil.

reduce interparticle contact areas, decrease attractive van der Waals interactions, and thus enabling well particle dispersion[6,22]. Nevertheless, the import of foreign chemical surfactants or nanoprotrusions inevitably shields a portion of the native particle surface, hence potentially influencing their intrinsic activities, especially after a complicated preparation process. Therefore, it is both fundamentally desirable and technically important to prepare particles that can be well dispersed in various liquids beyond the surface modification strategy.

Chemical heterogeneity has proved successful in tuning solid–liquid affinity, and achieved both superhydrophilicity and superoleophilicity on 2D plane surfaces, including TiO$_2$ and silicon surface[23,24]. For example, a 2D heterogeneous surface constituted by alternating hydrophilic and hydrophobic nanodomains can remarkably promote the wetting and spreading of both water and organic solvents by the capillary effect[24,25]. Inspired by the 2D heterogeneous surface, here we propose a surface heterogeneous nanostructuring strategy for the construction of omnidispersible particles, and we rationally expect that HL-HBPs exhibiting alternating hydrophilic domains and hydrophobic domains can realize well dispersion in both water and oil (Fig. 1b). The omnidispersible HL-HBPs, obtained without additional surface modification of chemical surfactants or nanoprotrusions, offer an unconventional approach for particle dispersion. As a proof-of-concept, we show the omnidipersible HL-HBPs enable a simultaneous recycle of organic dyes and regeneration of HL-HBPs from the synthetic wastewater merely through solvent exchange. The unique omnidispersity of HL-HBPs enables a good future for water purification and organic dye recycle as the next-generation separation materials.

## Results

### Synthesis and dispersion performance of the HL-HBPs

The HL-HBPs were synthesized by an emulsion interfacial polymerization approach (see Methods), as we have developed previously[26–28]. In a typical process for the synthesis of HL-HBPs, an oil-in-water emulsion was prepared, which contained hydrophilic monomers (sodium 4-styrenesulfonate, SS) in water and hydrophobic monomers (styrene, St; divinyl benzene, DVB) in oil. As shown in the scanning electron microscopy (SEM) images, the resultant poly(sodium 4-styrenesulfonate)-poly(styrene-divinyl benzene) (PSS-PSDVB) particles have uniform size

and porous structure (Fig. 2a, b). Their average diameter is 3.87 ± 0.14 μm (Mean ± SD, $n > 200$), and pore size ranges from several nanometers to hundreds of nanometers (Supplementary Fig. 1). Two domains are clearly verified by the cross-section transmission electron microscopy (TEM) images, the outer domain indicates hydrophilic PSS and the inner domain indicates hydrophobic PSDVB (Fig. 2c, d). Due to the porous structure, the hydrophilic domains and hydrophobic domains are both exposed to the surroundings. Photo-induced force infrared spectra[29,30] obtained at the margin of a particle from the cross-section sample of PSS-PSDVB HL-HBPs exhibit varied characteristic absorption peaks at bands of 1102 cm$^{-1}$, 1491 cm$^{-1}$ (and 1451 cm$^{-1}$), and 1735 cm$^{-1}$ with positions from 1 to 5 (Fig. 2e), which could be attributed to PSS[31], PSDVB[27,32], and epoxy[33] (embedding substrates), respectively. PiFM images obtained at these bands further confirm the distribution of hydrophilic domains (PSS), hydrophobic domains (PSDVB), and embedding substrates (epoxy) (Fig. 2f, g). More intuitively, the distribution of the hydrophilic domains and hydrophobic domains was directly characterized on the particle surface by adhesion force difference with atomic force microscopy (AFM) under the PeakForce QNM mode[34,35]. As shown in the topographic images and adhesion force images, the adhesion force between the domains with lower height (PSDVB) and the AFM tip is higher than that between the domains with higher height (PSS) and the AFM tip (Fig. 2h–k), further indicating the heterogeneity on the particle surface. Emulsion interfacial polymerization is a generalized approach for the synthesis of HL-HBPs with varied parameters. HL-HBPs with positive/negative charge, tunable particle diameter, and controlled Brunauer-Emmett-Teller (BET) surface area, could be prepared by adjusting monomer type, diameter of polystyrene (PS) template particles, and volume of porogen, respectively (Supplementary Figs. 2–4).

### Dispersion performance

Subsequently, we investigated and compared the dispersion performance of HL-HBPs with hydrophilic particles (HLPs) and hydrophobic particles (HBPs) in various common solvents from high polarity to low polarity, including water, dimethyl sulfoxide (DMSO), acetonitrile (ACN), ethanol, ethyl acetate (EA), toluene, tetrachloromethane (TCM), and octane. The HL-HBPs can be dispersed in all the above solvents (Fig. 3a), implying their unique omnidispersity. In contrast, HLPs, such as sulfonated group-modified PSDVB particles (Supplementary Fig. 5a), can be well dispersed in solvents with relatively high polarity including water, DMSO, ACN, ethanol, and EA, but they tend to agglomerate in solvents with relatively low polarity including toluene, TCM, and octane (Fig. 3b). In addition, HBPs, such as PSDVB particles (Supplementary Fig. 5b), can be well dispersed in all organic solvents, but they tend to agglomerate in water (Fig. 3c). In some cases, large density difference compared with employed solvents may cause sedimentation or floatation of HL-HBPs, but they can be easily dispersed again just upon gentle shaking. Time-lapse contact angle measurement also indicates that the HL-HBPs show improved hydrophilicity compared with HBPs, and oil droplets can spread rapidly on a substrate adhered with HL-HBPs (Supplementary Fig. 6). All these results indicate superior omnidispersity performance of HL-HBPs for various solvents.

### Theoretical calculations for the dispersion mechanism of the HL-HBPs

To elucidate the unconventional dispersion property of the HL-HBPs, we calculated the interaction potential between two HL-HBPs using classical Derjaguin–Landau–Verwey–Overbeek (DLVO) theory (Fig. 3d–f). In typical cases of water and oil (octane), two HL-HBPs are separated by a distance $d$ (Fig. 2d). The pair potential of HL-HBPs ($V_{total}$) is attributed to the sum of van der Waals interaction potential ($V_{vdW}$) and electrostatic interaction potential ($V_{DL}$), i.e., $V_{total} = V_{vdW} + V_{DL}$. Firstly, the $V_{vdW}$ was calculated according to the Hamaker

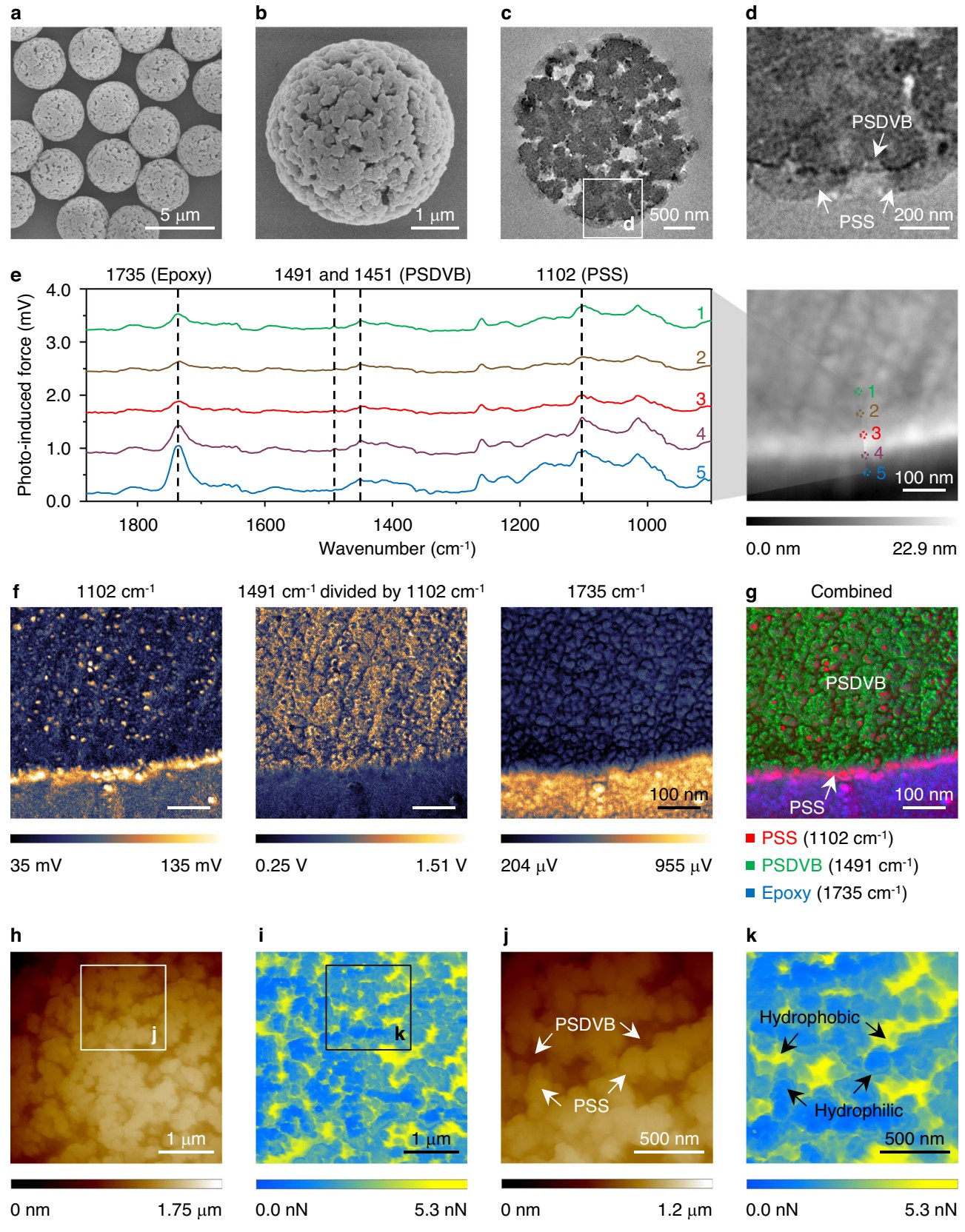

**Fig. 2 | Synthesis and characterizations of the HL-HBPs. a**, **b** SEM images of the HL-HBPs. **c**, **d** Cross-section TEM images of the HL-HBPs. Hydrophilic domains (PSS) and hydrophobic domains (PSDVB) are clearly verified. **e** Photo-induced force infrared spectra obtained at the margin of a particle from the cross-section sample of PSS-PSDVB HL-HBPs. **f** PiFM images obtained at different wavenumber. They reveal the distribution of PSS (1102 cm⁻¹), PSDVB (1491 cm⁻¹ divided by 1102 cm⁻¹), and epoxy (1735 cm⁻¹), respectively. **g** A combined image contrasts the distribution of PSS, PSDVB, and epoxy. **h**, **j** AFM topographic images and **i**, **k** corresponding adhesion force images of HL-HBPs.

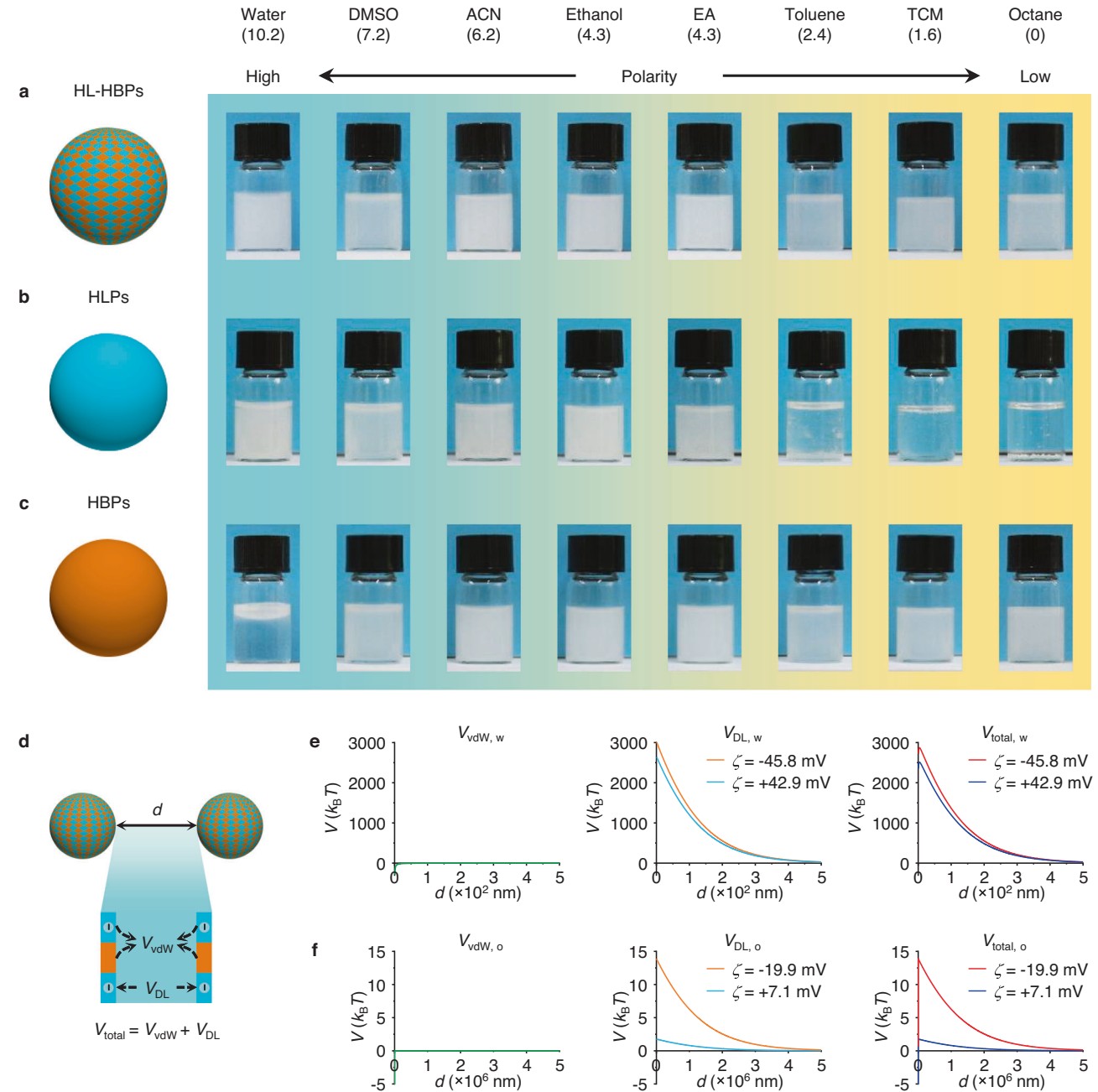

**Fig. 3 | Dispersion performance of the HL-HBPs. a** Dispersion photos of HL-HBPs in various solvents from high polarity to low polarity, including water, dimethyl sulfoxide (DMSO), acetonitrile (ACN), ethanol, ethyl acetate (EA), toluene, tetrachloromethane (TCM), and octane. **b** Dispersion photos of hydrophilic particles (HLPs). **c** Dispersion photos of hydrophobic particles (HBPs). **d** Scheme for interaction potential calculation. The total interaction potential ($V_{total}$) between two HL-HBPs separated by a distance $d$ is calculated according to attractive van der Waals interaction potential ($V_{vdW}$) and repulsive electrostatic interaction potential ($V_{DL}$). **e** $V_{vdW, w}$, $V_{DL, w}$, and $V_{total, w}$ represent interactions between two HL-HBPs in water. **f** $V_{vdW, o}$, $V_{DL, o}$, and $V_{total, o}$ represent interactions between two HL-HBPs in oil (octane). A positive $V_{total}$ indicates repulsive force and stable dispersity.

model[36], in which the HL-HBPs were treated as core-shell particles[37] (see Methods). It should be noted that the Hamaker constant of the HL-HBPs are also affected by the oil since oil can permeate into the pores of the particles (Supplementary Fig. 7). Secondly, the $V_{DL}$ resulting from the charged hydrophilic domains was calculated by the Poisson–Boltzmann equation[38]. Two HL-HBPs, including negatively charged PSS-PSDVB and positively charged poly(2-trimethylammoniumethyl methacrylate chloride)-PSDVB (PTMAEMC-PSDVB), were calculated for comparison. The results show that the $V_{total, w}$ of HL-HBPs in water is positive at $d > 0$ and reaches maximum ~2873 $k_BT$ for negatively charged PSS-PSDVB HL-HBPs ($\zeta = -45.8$ mV), and ~2511 $k_BT$ for positively charged PTMAEMC-PSDVB HL-HBPs ($\zeta = +42.9$ mV)

(Fig. 2e), suggesting the dispersion of HL-HBPs in water is quite stable. In octane, $V_{vdW, o}$ and $V_{DL, o}$ are much reduced compared with those in water, and $V_{DL, o}$ is not screened in a long range, resulting in a maximum $V_{total, o}$ of ~14 $k_BT$ for negatively charged PSS-PSDVB HL-HBPs ($\zeta = -19.9$ mV), and ~2 $k_BT$ for positively charged PTMAEMC-PSDVB HL-HBPs ($\zeta = +7.1$ mV) (Fig. 2f). These results indicate that the dispersion of HL-HBPs in oil is stable. The omnidispersity of HL-HBPs can be attributed to the unique hydrophilic-hydrophobic heterostructure. First, the hydrophilic domains are favorable for the wetting of water, and the hydrophobic domains are favorable for the wetting of oil, making it easy for both water and oil to fill the gaps between particles. Second, introduction of charge on hydrophilic domains offers interparticle

repulsive force, preventing the particles from aggregating. Therefore, the unique HL-HBPs are dispersible in both water and oil.

The surface heterogeneous nanostructuring strategy demonstrates superior advantages. First, it avoids the import of foreigner organic surfactants or nanostructures, especially those obtained after a complicated process. Second, it achieves omnidispersion in various solvents with wide-range polarity, which is only realized with rare examples of existing materials. Even with these advantages, this strategy is only proved by the emulsion interfacial polymerization in functional particle synthesis. Its generality and uniqueness remain to be explored in other functional material systems.

## Dye adsorption performance

Organic dyes are popular additives that can make industrial products and household commodities colorful[39], and can be used to visualize and interpret biological events[40]. The global production of organic dyes approaches 700000 tons annually, however, nearly 10–15% of them is discharged into industrial and household wastewater, which has become an important source of water pollution and a non-negligible threat for public health[41–44]. To treat dyed wastewater, most existing strategies are mainly concentrated on the removal of dyes from water. The conventional coagulation-flocculation method can remove part of the organic dyes from wastewater, after which the dye-containing sludge is always dumped[45,46]. In addition, this approach is mainly suitable for water-insoluble dyes, but generally not suitable for water soluble dyes. Alternatively, biological degradation uses biological materials, such as algae, bacteria, fungi, and yeasts, to disintegrate organic dyes. This approach also generates biomass sludge[47]. State-of-the-art methods seek to remove organic dyes from water maximumly and more sustainably, such as catalytic oxidation[48], membrane nanofiltration[49,50], and porous particle adsorption[51,52]. Organic dyes are degraded into smaller molecules, excluded from water, or extracted from water. These catalysts, membranes, and adsorbent materials can be repeatedly used after desorption or regeneration. Nevertheless, the recycling of organic dyes has received rare attention. A series of porous materials, such as activated carbons[53–55], metal-organic frameworks (MOFs)[51,56,57], networks[52], and layered double hydroxides (LDHs)[58–60], have been developed for dye adsorption and desorption. To desorb organic dyes, commonly used eluents containing inorganic acid, alkaline, or salt, are often added into the aqueous solution to weaken the interactions between materials and dyes[58,61,62]. However, such eluents result in secondary dye wastewater, and render the recycle process of organic dyes more complicated. The HL-HBPs, taking advantages of omnidispersity, show great promise for organic dye separation and recycle.

First, we verified the adsorption performance of HL-HBPs for organic dyes with opposite charge from wastewater (Fig. 4). For example, cationic dyes are adsorbed and separated by HL-HBPs with negatively charged hydrophilic domains (Fig. 4a). Anionic dyes are adsorbed and separated by HL-HBPs with positively charged hydrophilic domains (Fig. 4b). Typically, negatively charged PSS-PSDVB HL-HBPs with an average zeta potential ($\zeta$) of −45.8 mV (Supplementary Fig. 2a, c) were used for the adsorption of cationic dyes, including malachite green (MG), methyl violet 2B (MV), rhodamine B (RB), crystal violet (CV), and methylene blue (MLB). The maximum adsorption amounts per unit mass of the HL-HBPs for MG, MV, RB, CV, and MLB are $13.62 \pm 0.05 \ \mu g \ mg^{-1}$, $7.00 \pm 0.04 \ \mu g \ mg^{-1}$, $6.02 \pm 0.03 \ \mu g \ mg^{-1}$, $5.92 \pm 0.04 \ \mu g \ mg^{-1}$, and $4.64 \pm 0.04 \ \mu g \ mg^{-1}$ (Mean ± SD, $n = 3$), respectively (Fig. 4c). In addition, positively charged PTMAEMC-PSDVB HL-HBPs with an average $\zeta$ of +42.9 mV (Supplementary Fig. 2b, d) were used for the adsorption of anionic dyes, including methyl blue (MB), acid fuchsin (AF), Congo red (CR), Evans blue (EB), and methyl orange (MO). The maximum adsorption amounts per unit mass of the HL-HBPs for MB, AF, CR, EB, and MO are $11.50 \pm 0.14 \ \mu g \ mg^{-1}$, $5.68 \pm 0.14 \ \mu g \ mg^{-1}$, $4.51 \pm 0.18 \ \mu g \ mg^{-1}$, $1.48 \pm 0.21 \ \mu g \ mg^{-1}$, and $1.25 \pm 0.15 \ \mu g \ mg^{-1}$

(Mean ± SD, $n = 3$), respectively (Fig. 4d). We noted that all organic dyes can be adsorbed by HL-HBPs as expected, but there is much difference between the adsorption amounts. It is difficult to explain such difference merely by electrostatic interaction. The difference in adsorption amounts for various organic dyes could be attributed to their intrinsic interaction site, including the electrostatic interaction site, hydrogen bonding interaction site, and hydrophobic/π-π bonding interaction site[63]. To better understand the difference, we attempt to utilize an adsorption index by calculating the ratio of the total number of interaction site to molecule weight of the dyes ($n_{Sum}/M_W$) to compare the interaction site of different dyes with same mass (Fig. 4e, f, and Supplementary Table 3). Calculation results indicate that the adsorption index is largely positive correlation with the dye adsorption amount. RB shows a slight deviation for the correlation between adsorption amount and adsorption index. Such deviation may occur between dyes with similar structures like MV, RB, and CV, probably because some of the groups are not taken into consideration when calculating the adsorption index, such as methyl groups and ethyl groups. These results suggest $n_{Sum}/M_W$ is a reasonable adsorption index to explain the adsorption amount difference between various organic dyes. It should also be noted that the interaction energy of electrostatic interaction (-100 $kT$) is much higher than that of other interactions, including hydrogen bonding interaction (5–10 $kT$) and hydrophobic/π-π bonding interaction (-1 $kT$)[8]. Nevertheless, it remains a difficult task to clearly and quantitatively define the contribution factor of different types of interactions for dye adsorption due to the unique heterostructure. The adsorption index is an immature attempt to explain the difference in dye adsorption amounts. Further modification is expected for more accurate explanations and prediction as the development of experimental techniques and fundamental theories. Therefore, the HL-HBPs are effective for the adsorption of various organic dyes from wastewater, with the synergy of electrostatic interaction, hydrogen bonding interaction, and hydrophobic/π-π bonding interaction.

## Dye adsorption kinetics and efficiency

The dye adsorption kinetics of HL-HBPs was investigated by in situ laser scanning confocal microscopy (LSCM) and adsorption amount measurement experiment. We tested the adsorption of a cationic fluorescent organic dye, RB, on negatively charged PSS-PSDVB HL-HBPs as an example. LSCM images show that RB can be significantly adsorbed onto PSS-PSDVB HL-HBPs within 5 s, and there is no apparent increase within 3 min (Fig. 5a). Quantificationally, the dye adsorption efficiency was tested for kinetics study when the concentration of HL-HBPs varied from $1.33 \ mg \ mL^{-1}$ to $3.33 \ mg \ mL^{-1}$, $6.67 \ mg \ mL^{-1}$, $10 \ mg \ mL^{-1}$, and $13.33 \ mg \ mL^{-1}$. The adsorption equilibrium was almost achieved at adsorption time of 5 s, and adsorption efficiency rarely increased in 5 min (Fig. 5b). The rapid dye adsorption equilibrium of HL-HBPs indicates their capability of dye adsorption, which is difficult to achieve with existing adsorbent materials, such as LDHs (from 5 min to tens of minutes)[58–60], activated carbons (at least tens of minutes)[53–55], and MOFs (tens of hours)[51,57] (Supplementary Table 4).

We further investigated the influence of particle diameter and BET surface area of HL-HBPs on the adsorption of organic dyes. Four HL-HBPs, with particle diameter of $3.87 \pm 0.14 \ \mu m$, $4.34 \pm 0.10 \ \mu m$, $6.20 \pm 0.67 \ \mu m$, and $12.56 \pm 0.83 \ \mu m$ (Mean ± SD, $n > 200$, Supplementary Fig. 3), were compared for RB adsorption. For respective HL-HBPs, LSCM images obtained at adsorption time of 5 s and 3 min exhibit little difference, implying there is no obvious influence of particle diameter on their capability of rapid dye adsorption (Fig. 5c). In addition, the dyes can be adsorbed onto the innermost region of the HL-HBPs with diameter of 3.87 μm and 4.34 μm. However, the dye adsorption depth ($\eta$) for HL-HBPs with diameter of 6.20 μm and 12.56 μm is about 1.4 μm and 1.6 μm, respectively, suggesting that further increasing particle diameter has no obvious

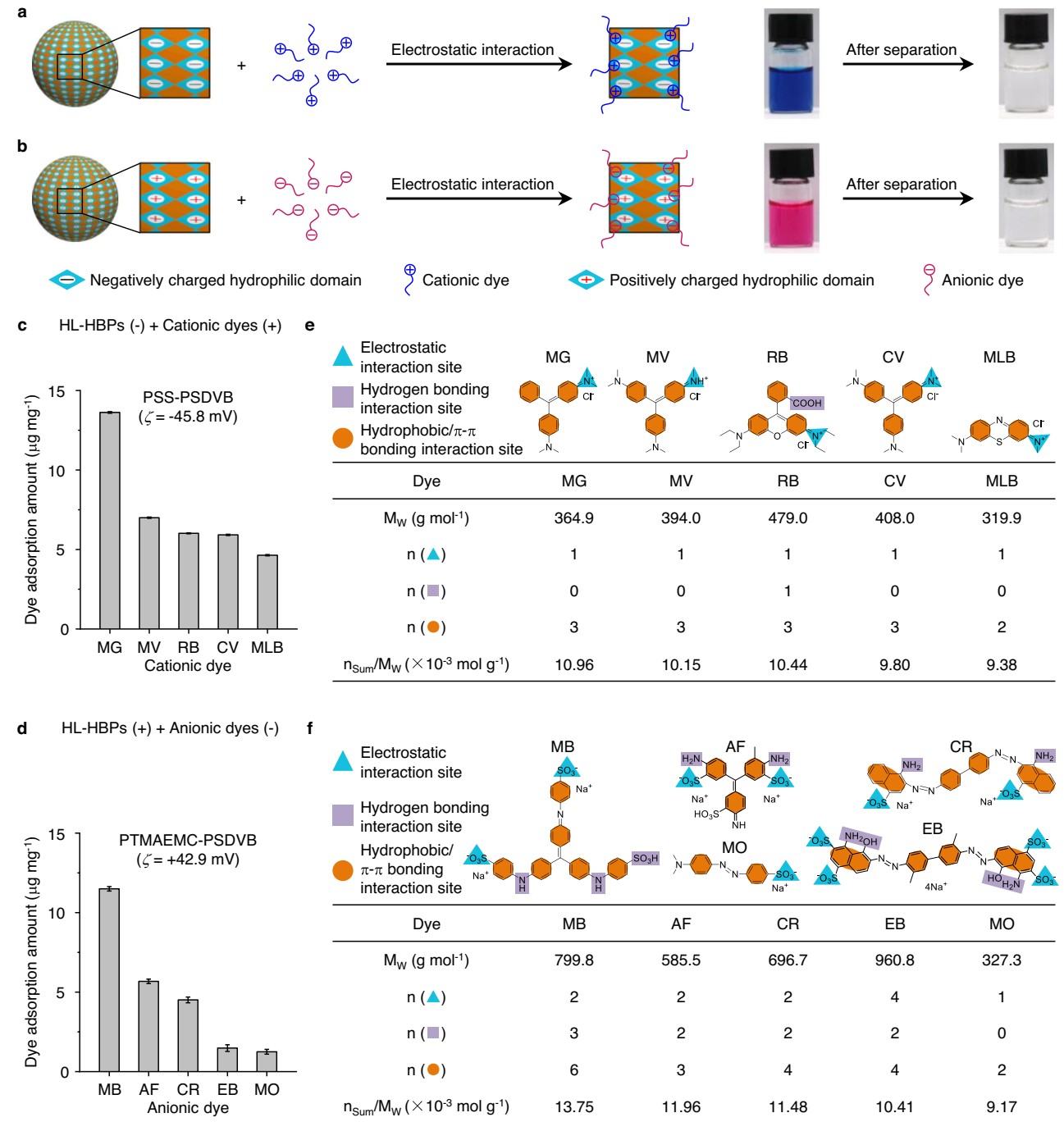

**Fig. 4 | Dye adsorption performance of HL-HBPs. a** HL-HBPs with negatively charged hydrophilic domains are used for the separation of cationic dyes. **b** HL-HBPs with positively charged hydrophilic domains are used for the separation of anionic dyes. **c** Adsorption amounts of various cationic dyes with negatively charged HL-HBPs (PSS-PSDVB). The initial concentration of HL-HBPs suspension and dye solution are 1.33 mg mL$^{-1}$ and 20 ppm, respectively. **d** Adsorption amounts of various anionic dyes with positively charged HL-HBPs (PTMAEMC-PSDVB). The initial concentration of HL-HBPs suspension and dye solution are 1.33 mg mL$^{-1}$ and 20 ppm, respectively. Dye adsorption efficiency, Mean ± SD, $n = 3$. **e** Calculation of interaction sites of cationic dyes. **f** Calculation of interaction sites of anionic dyes. $M_W$: molecule weight of the dyes. $n$ (▲): number of electrostatic interaction site per dye molecule. $n$ (■): number of hydrogen bonding interaction site per dye molecule. $n$ (●): number of hydrophobic/π-π bonding interaction site per dye molecule. $n_{Sum} = n$ (▲) $+ n$ (■) $+ n$ (●). $n_{Sum}/M_W$: total number of interaction site to molecule weight of the dyes.

effects on promoting the adsorption of dyes. Except for rapid adsorption, the organic dyes can be rapidly desorbed from HL-HBPs after adding the organic solvent (Supplementary Fig. 9). Additionally, four HL-HBPs, with BET surface area of 7.6 ± 0.1 m$^2$ g$^{-1}$, 10.2 ± 0.1 m$^2$ g$^{-1}$, 13.0 ± 0.1 m$^2$ g$^{-1}$, and 14.8 ± 0.1 m$^2$ g$^{-1}$ (Mean ± SD, $n = 3$) (Supplementary Fig. 4), were compared for the RB adsorption amount. These four HL-HBPs show an increased dye adsorption

amounts from 3.44 ± 0.04 μg mg$^{-1}$ to 3.83 ± 0.06 μg mg$^{-1}$, 4.46 ± 0.05 μg mg$^{-1}$, and 5.56 ± 0.06 μg mg$^{-1}$ (Mean ± SD, $n = 3$), as the BET surface area increases (Fig. 5d). The dye adsorption efficiency maintains >92% at pH range of 1–14 without significant decrement, and slightly decreases to 86.15 ± 1.09% (Mean ± SD, $n = 3$) when pH = 0 (Supplementary Fig. 10), indicating the dye adsorption can be achieved in a wide range of pH value. These

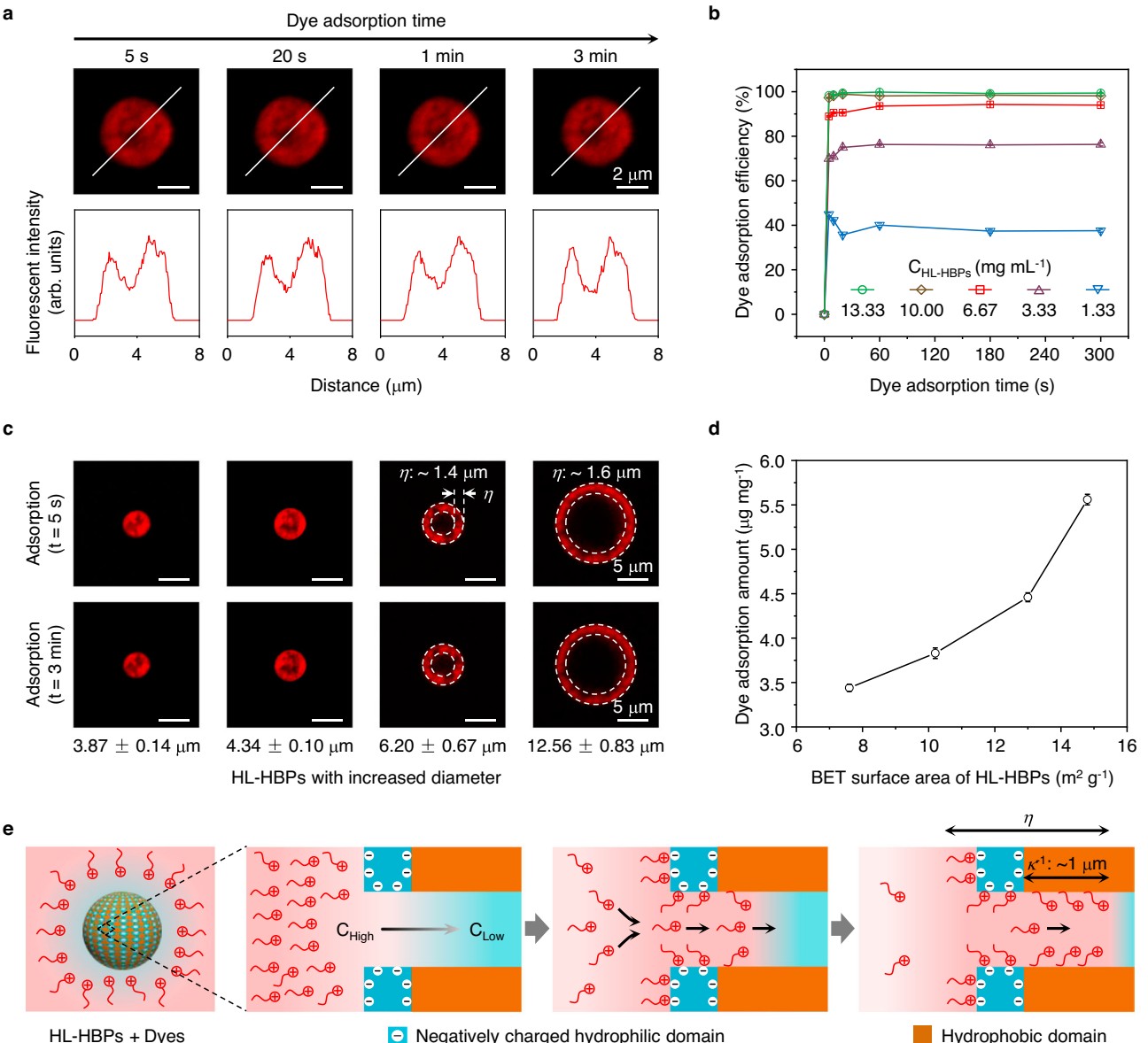

**Fig. 5 | Dye adsorption kinetics and mechanism with the HL-HBPs. a** In situ LSCM images and corresponding fluorescent intensity profiles of PSS-PSDVB HL-HBPs adsorbed with RB at different adsorption time. **b** RB adsorption efficiency with PSS-PSDVB HL-HBPs of various particle concentration ($C_{HL-HBPs}$) at different adsorption time. The initial concentration of RB solution is 20 ppm. Dye adsorption efficiency, Mean ± SD, $n = 3$. **c** LSCM images showing the adsorption of dyes on HL-HBPs with varied diameter. $\eta$: dye adsorption depth. Diameter: Mean ± SD, $n > 200$. **d** Dye adsorption amounts of HL-HBPs with varied Brunauer−Emmett−Teller (BET) surface area. The initial concentration of HL-HBPs suspension and RB solution are 1.33 mg mL$^{-1}$ and 20 ppm, respectively. Dye adsorption amount, Mean ± SD, $n = 3$. **e** Schematic of the dye adsorption process using HL-HBPs. A dye concentration gradient is formed from bulk solution ($C_{High}$) to particle interior ($C_{Low}$). Positively charged dyes can be adsorbed on the negatively charged hydrophilic domains (PSS) via electrostatic interaction and hydrogen bonding interaction, and be further attracted along the nanopores towards hydrophobic domains in the presence of electrostatic interaction and under the dye concentration gradient. Dyes can be adsorbed on hydrophobic domains via hydrophobic /π-π bonding interaction. The dye adsorption depth ($\eta$) is probably highly dependent on the Debye length ($\kappa^{-1}$).

results illustrate that the HL-HBPs show excellent capability of rapid dye adsorption and desorption.

The capability of rapid dye adsorption and desorption with HL-HBPs originates from their unique hydrophilic-hydrophobic surface heterogeneity with the charged surface. For RB adsorption with PSS-PSDVB HL-HBPs, a dye concentration gradient is formed from bulk solution ($C_{High}$) to particle interior ($C_{Low}$) (Fig. 5e, left). Positively charged dyes (RB) can be adsorbed on the negatively charged hydrophilic domains (PSS) via electrostatic interaction and hydrogen bonding interaction, and be attracted along the nanopores towards hydrophobic domains (PSDVB) in the presence of electrostatic interaction and under the dye concentration gradient (Fig. 5e, middle). Subsequently, the dyes can be adsorbed on hydrophobic domains via

hydrophobic/π-π bonding interaction (Fig. 5e, right). Dye adsorption depth ($\eta$) is probably highly dependent on the Debye length ($\kappa^{-1}$), which is used to indicate the working range of electrostatic interaction. In water, $\kappa^{-1}$ is about 1 μm[8]. Further considering hydrogen bonding interaction and hydrophobic/π-π bonding interaction, it is reasonable that total depth for dye adsorption in the hydrophilic regions and hydrophobic regions of HL-HBPs approximates 1.4–1.6 μm (Fig. 5c and Fig. 5e, right). The outer hydrophilic domains are composed of negatively charged PSS with a thickness of tens of nanometers, which provide intense electrostatic field for strong electrostatic interaction (Supplementary Fig. 11). Positively charged organic dyes in a long range (~1 μm) can be attracted immediately to neutralize the charge of hydrophilic domains. Upon attracted towards the HL-HBPs, the

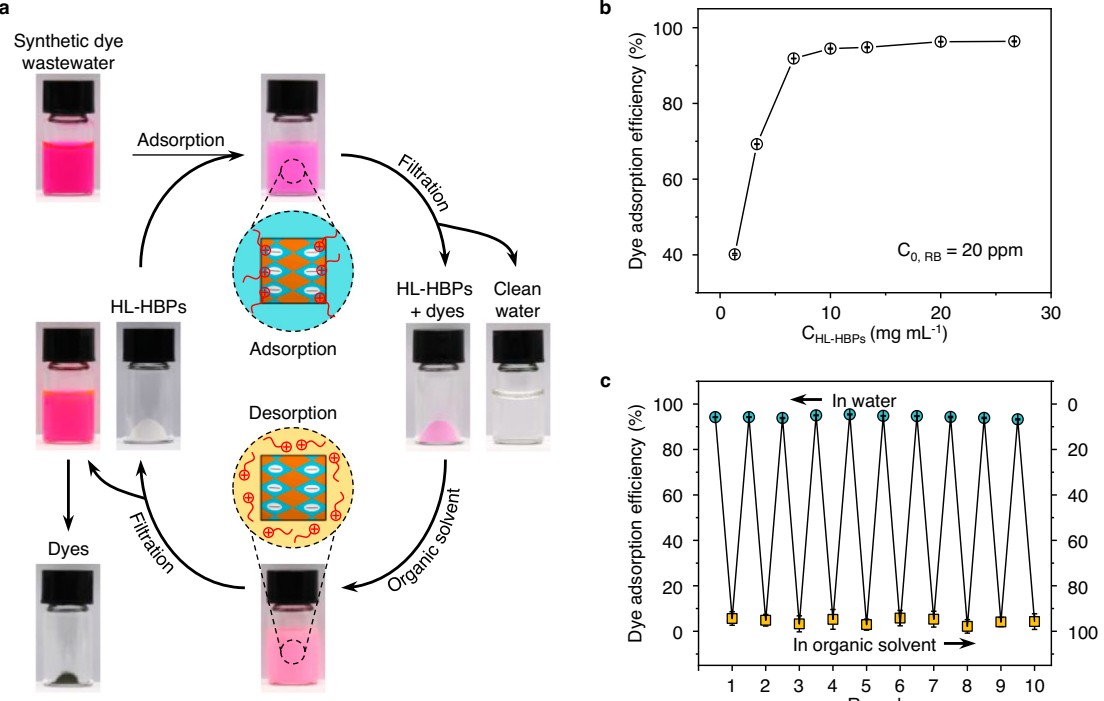

**Fig. 6 | Omnidispersity-dependent solvent exchange assisted dye recycle using the HL-HBPs. a** Schemes and photos for the recycle of RB using PSS-PSDVB HL-HBPs. First, the dyes were adsorbed by HL-HBPs in aqueous solution. Second, HL-HBPs adsorbed with dyes and clean water were obtained by filtration. Third, the dyes were desorbed from HL-HBPs by adding organic solvents. Fourth, HL-HBPs and dyed organic solution were obtained by filtration. Finally, dye powders were obtained by distillation, and HL-HBPs can be used for repeated dye recycle. **b** Relationship between the concentration of PSS-PSDVB HL-HBPs ($C_{HL-HBPs}$) and dye (RB) adsorption efficiency. The initial dye concentration ($C_{0, RB}$) is 20 ppm. Dye adsorption efficiency, Mean ± SD, $n = 3$. **c** Dye adsorption efficiency in water and desorption efficiency in organic solvents using HL-HBPs at different recycle rounds. Dye adsorption and desorption efficiency, Mean ± SD, $n = 3$.

hydrophobic domains on HL-HBPs provide short-range hydrophobic/π-π bonding interaction. Most of the pores exhibit pore diameter from 10 nanometers to 100 nanometers (Supplementary Fig. 4c, d), which allows rapid diffusion and adsorption of organic dye molecules. When the surface charge of HL-HBPs is neutralized, the diffusion of organic dyes towards the particles is almost halted, and adsorption equilibrium is achieved. For the desorption process, water was replaced by organic solvent, these interactions were significantly weakened, and organic dyes were rapidly desorbed from both hydrophilic domains and hydrophobic domains and released into organic solvent.

**Solvent exchange-assisted dye recycle using the HL-HBPs**

In a proof-of-concept, we demonstrate the capability of the HL-HBPs to recycle organic dyes from synthetic dye wastewater by taking advantage of their omnidispersity. The recycle of cationic dye, RB, using PSS-PSDVB HL-HBPs was shown as an example (Fig. 6a). First, the synthetic dye wastewater was mixed with the HL-HBPs aqueous suspension to achieve dye adsorption. Second, the particles and the aqueous solution were separated by filtration, resulting in dyed particles and clean water. Third, the HL-HBPs adsorbed with dyes were dispersed in organic solvent to achieve dye desorption. Fourth, dyed organic solution and white particle powders were separated after filtration. Finally, the organic solution was distilled to recycle the dye powders. In the dye adsorption process, the concentration of HL-HBPs should be large enough to realize high dye adsorption efficiency. We found that the adsorption efficiency increases as $C_{HL-HBPs}$ increases, and hits a plateau at 10 mg mL$^{-1}$ (Fig. 6b). Therefore, we used 10 mg mL$^{-1}$ of HL-HBPs for dye adsorption. To desorb dyes adsorbed on the HL-HBPs, many potential organic solvents can be chosen according to the following principles: (1) dyes can be dissolved in the organic solvents, (2) the boiling point of the organic solvents should be as low as possible to

facilitate dye recycle, and (3) the organic solvents can weaken the interactions between dyes and HL-HBPs. Following these principles, a series of organic solvents with low boiling point and varied polarity can be used, such as ACN, ethanol, EA, TCM, *n*-hexane, tetrahydrofuran (THF), methanol, acetone, dichloromethane (DCM), *n*-pentane, and ethyl ether (EE) (Supplementary Table 5). To desorb RB from PSS-PSDVB HL-HBPs, DCM/methanol/ethanol, or their mixed solution with *n*-pentane/*n*-hexane, can be used as the eluent. The organic solvent was distilled at a lower temperature than the boiling point of water, for example, 80 °C for the mixed solution of ethanol and *n*-hexane (1:1 v/v), or 40 °C for the mixed solution of DCM and *n*-pentane (1:1 v/v). After distillation, dye powders were recycled. Furthermore, the HL-HBPs can be used for next round dye adsorption and desorption. In 10 dye recycle rounds, the dye adsorption efficiency and desorption efficiency maintain 93–96% and 94–98%, respectively (Fig. 6c). The regenerated HL-HBPs have rarely changed after 10 rounds of adsorption−desorption process, and there are rare additional impurities in the recycled dyes (Supplementary Figs. 12, 3). Minority of the HL-HBPs can be retained on the membrane during the filtration process (Supplementary Fig. 14), and the regeneration ratio of the HL-HBPs is about 95.5%. After one round of adsorption−desorption process, the recycle ratio of the dyes is about 92.2%. By using a reactor with a capacity of 5 L, the synthesis of HL-HBPs can be scaled up, thus the dye recycle strategy could also be scaled up from several milliliters to 2 L (Supplementary Figs. 15, 16). The HL-HBPs provide a promising candidate for organic dye recycle compared with existing materials. To desorb organic dyes from materials reported in literatures, eluents containing inorganic acid, alkaline, or salt, are often added into the aqueous solution to weaken the interactions between materials and dyes (Supplementary Table 6). There is unique significance to use

organic solvents instead of inorganic solvent for organic dye recycle. Organic solvents are easy to be removed from dyes by simple distillation. In contrast, additional steps are required to remove inorganic acid, alkaline, and salt from aqueous solution of recycled dyes.

## Discussion

In summary, we propose a unique surface heterogeneous nanostructuring strategy for particle dispersion, and synthesize a series of HL-HBPs exhibiting superior omnidispersity in various solvents from water to oil with different polarity and boiling point. TEM, PiFM, and adhesion force images all reveal the alternating hydrophilic domains and hydrophobic domains of the HL-HBPs. The surface heterogeneity allows HL-HBPs to be dispersed in different solvents from high-polarity water to low-polarity oil. Benefiting from their omnidispersity, the HL-HBPs demonstrate a repeatable and efficient recycle of organic dyes from synthetic wastewater samples merely using solvent exchange. Our HL-HBPs suggest remarkable advances in both scientific understanding and technological innovation. Although the dye adsorption amount of HL-HBPs is lower than those existing materials with high surface area like MOFs, their adsorption rate is rapid. The next challenge is to create adsorbent materials with high adsorption capacity while maintaining the rapid adsorption performance. These HL-HBPs are expected for extensive separation of various contaminants from polluted water, production of clean water, recycle of dumped resources, and will boost the development of the ecological environment.

## Methods

### Abbreviations

Important abbreviations and corresponding full names were listed in Supplementary Table 1.

### Synthesis of negatively charged PSS-PSDVB HL-HBPs

Firstly, 1-chlorododecane (0.1 mL) was emulsified in sodium dodecyl sulfate (SDS) aqueous solution (0.25% w/w, 10 mL), and uniform PS particles (200 mg) were dispersed in SDS aqueous solution (0.25% w/w, 20 mL) under ultrasonication. The emulsion and suspension were successively poured into a glass flask, magnetically stirred and maintained at 40 °C for 20 h. Secondly, SS (500 mg) was dissolved in SDS aqueous solution (0.25% w/w, 10 mL), and then mixed with St (2 mL), DVB (1 mL), and 2,2′-azoisobutyronitrile (40 mg). The mixed solution was used for ultrasonication to prepare an oil-in-water emulsion. The emulsion was poured into the glass flask and stirred at 40 °C for another 6 h. Thirdly, polyvinyl alcohol (PVA) solution (1% w/w, 5 mL) was added into the glass flask, then the solution was deoxygenated with nitrogen for 5 min, followed by which the temperature was raised to 70 °C and maintained for at least 10 h. Finally, the resultant products were thoroughly washed with deionized water and ethanol.

### Synthesis of positively charged PTMAEMC-PSDVB HL-HBPs

Firstly, 1-chlorododecane (0.1 mL) was emulsified in PVA aqueous solution (1% w/w, 10 mL), and uniform PS particles (200 mg) were dispersed in PVA aqueous solution (1% w/w, 10 mL) under ultrasonication. The emulsion and suspension were successively poured into a glass flask, magnetically stirred and maintained at 40 °C for 20 h. Secondly, TMAEMC aqueous solution (75 wt% in water, 1334 mg) was added in PVA aqueous solution (1% w/w, 10 mL), and then mixed with St (3 mL), DVB (1 mL), toluene (0.6 mL), and 2,2′-azoisobutyronitrile (40 mg). The mixed solution was used for ultrasonication to prepare an oil-in-water emulsion. The emulsion was poured into the glass flask and stirred at 40 °C for another 6 h. Thirdly, PVA solution (1% w/w, 5 mL) was added into the glass flask, then the solution was deoxygenated with nitrogen for 5 min, followed by which the temperature was raised to 70 °C and maintained for at least 10 h. Finally, the

resultant products were thoroughly washed with deionized water and ethanol.

### Post-treatment of HL-HBPs

The pore interconnectivity of the synthesized HL-HBPs can be additionally improved by dissolving and removing the linear PS, which was introduced as template particles during the synthesis process of HL-HBPs. Typically, HL-HBPs were dispersed in DCM for 1 h, and then isolated from DCM by centrifugation. This process was repeated for three times. Finally, the resultant particles were thoroughly dried at ambient temperature.

### Electron microscopy

SEM (Hitachi, SU8010) and TEM (FEI Tecnai, $G^2$ 20) were used to characterize the morphology of the particles. Before SEM observation, a thin layer of golden nanoparticles was deposited on the particle surface. Before TEM observation, the particles were embedded in epoxy resin (SPI-PON 812), cured at 60 °C for 24 h, cut into thin sections (thickness: ~100 nm). In addition, the sections obtained from PTMAEMC-PSDVB HL-HBPs were stained with phosphotungstic acid solution (2 wt% in water, pH = 6.5, ordered from Zhongjingkeyi (China)) for 2 min.

### PiFM infrared spectra and imaging

PiFM is a multimodal AFM technique that combines AFM and spectroscopy to obtain nondestructive topographic and absorption information with sub ~10 nm resolution simultaneously via tip-enhanced near-field imaging and spectroscopy[29,30]. Before PiFM characterization, the HL-HBPs were embedded in epoxy resin (SPI-PON 812), cured at 60 °C for 24 h, cut into thin sections (thickness: ~100 nm), and deposited on a flat silicon wafer. PiFM spectra were taken with a pitch of 16 nm, an acquisition time of 20 s and were power normalized. For spectral acquisition, the laser sweeps through its full range with a dwell time of ~17 ms/cm⁻¹ (spectral range/time per spectrum), during which the probe records the PiFM response at the region of interest with sub ~10 nm resolution. For PiFM imaging, the tip was illuminated by a tunable coherent infrared source at 1102 cm⁻¹, 1451 cm⁻¹, 1491 cm⁻¹, and 1735 cm⁻¹. Images were acquired at scan speeds of 0.89 Hz (500 nm² images) and 0.39 Hz (4.14 μm² images) and a resolution of 256 × 256 pixels. All PiFM measurements were carried out using a VistaScope microscope from Molecular Vista Inc. (San Jose, CA, USA), coupled with Block Engineering's LaserTune QCL, with a range of 753 cm⁻¹ to 1905 cm⁻¹ and a spectral line width of 2 cm⁻¹. Gold-coated NCH 300 kHz non-contact cantilevers from Nanosensors were used for all measurements. All images and data processing were carried out using SurfaceWorks.

### AFM

Topographic images and adhesion force images were obtained on an AFM (Dimension FastScan Bio-ICON, Bruker). A thin layer of polydimethylsiloxane (PDMS) precursor was spin coated on a clean silicon wafer, and then was partially cured at 80 °C for 40 min. Subsequently, the HL-HBPs were dispersed in ethanol, and spin coated on the PDMS-silicon wafer, after which the PDMS was completely cured to immobilize the particles. For AFM observation, images with 512 × 512 pixels were obtained using the mode of Quantitative Nanomechanical Mapping/PeakForce QNM in Air. The employed AFM tip is silicon tip on a nitride triangular cantilever (SCAN ASYST-AIR, Bruker), and the spring constant is 0.4 N m⁻¹. The load force applied in the experiment is ~1.6 nN and scan rate is ~0.5 Hz.

### Theoretical calculations for the dispersion mechanism

Calculations of $V_{vdW}$. The $V_{vdW}$ was calculated by the Hamaker model for the cases of water and octane, respectively. To simplify calculation, the HL-HBPs are treated as core-shell particles. Two HL-HBPs of radius

$R$ and shell thickness of $\delta$ are separated by a distance $d$. The Hamaker constants of the core, the shell and the medium solvent are $A_c$, $A_s$, and $A_m$, respectively. We assume the pores of HL-HBPs are filled with water when they are dispersed in water, while filled with oil when they are dispersed in oil (Supplementary Fig. 7), and the corresponding Hamaker constants are given by[37]

$$A_C = \left((1-p)A_{PSDVB}^{1/2} + pA_m^{1/2}\right)^2 \tag{1}$$

$$A_S = \left((1-p)A_{PSS}^{1/2} + pA_m^{1/2}\right)^2 \tag{2}$$

where $p$ represents the particle porosity, which is obtained from the pore volume when the particle density is assumed to be $1\,g\,cm^{-3}$. Here the Hamaker constant of pure substance 1 is calculated by[8]

$$A = \frac{3}{4}k_B T\left(\frac{\varepsilon_1 - \varepsilon_3}{\varepsilon_1 + \varepsilon_3}\right)^2 + \frac{3h\nu_e}{16\sqrt{2}}\left(\frac{(n_1^2 - n_3^2)^2}{(n_1^2 + n_3^2)^{3/2}}\right) \tag{3}$$

where $k_B = 1.38 \times 10^{-23}\,J\,K^{-1}$, $T = 300\,K$, $h = 6.62 \times 10^{-34}\,J\,s$, $\nu_e = 3 \times 10^{15}\,s^{-1}$, $\varepsilon_3 = 1$, and $n_3 = 1$ for vacuum[37]. The values used for calculation are shown in Supplementary Table 2.

The $V_{vdW}$ of the HL-HBPs are given by[37]

$$V_{vdW} = -\frac{1}{12}\left[H_{SS}\left(A_s^{1/2} - A_m^{1/2}\right)^2 + H_{CC}\left(A_c^{1/2} - A_s^{1/2}\right)^2 + 2H_{CS}\left(A_c^{1/2} - A_s^{1/2}\right)\left(A_s^{1/2} - A_m^{1/2}\right)\right] \tag{4}$$

where the unretarded $H$ functions are given by

$$H(x,y) = \frac{y}{x^2 + xy + x} + \frac{y}{x^2 + xy + x + y} + 2\ln\left[\frac{x^2 + xy + x}{x^2 + xy + x + y}\right] \tag{5}$$

$$x = \frac{\Delta}{2r_1}, y = r_2/r_1 \tag{6}$$

$$H_{SS}: \Delta = d, r_1 = r_2 = R \tag{7}$$

$$H_{CC}: \Delta = d + 2\delta, r_1 = r_2 = R - \delta \tag{8}$$

$$H_{CS}: \Delta = d + \delta, r_1 = R, r_2 = R - \delta \tag{9}$$

Calculations of $V_{DL}$. The $V_{DL}$ is calculated by the Poisson–Boltzmann equation

$$V_{DL} = 2\pi\varepsilon_0\varepsilon_r R\zeta^2\ln\left(1 + e^{-\kappa d}\right) \tag{10}$$

where $\varepsilon_0 = 8.85 \times 10^{-12}$ Farad $m^{-1}$, $\varepsilon_r$ is the dielectric constant of the solvent, $\zeta$ is the zeta potential of the particles in the solvent, $\kappa^{-1}$ is the Debye length.

For HL-HBPs dispersed in water, we used $\varepsilon_r = 80.2$, $\kappa^{-1} = 100\,nm$. In addition, $\zeta$ (PSS-PSDVB) = $-45.8\,mV$, $\zeta$ (PTMAEMC-PSDVB) = $+42.9\,mV$.

For HL-HBPs dispersed in octane, we used $\varepsilon_r = 1.948$. To simplify the calculation process, we assumed $\kappa^{-1} = 100000\,nm$ since the Debye length dramatically increases owing to the weak ionic shielding effect in solvents with low dielectric constant[64]. In addition, $\zeta$ (PSS-PSDVB) = $-19.9\,mV$, $\zeta$ (PTMAEMC-PSDVB) = $+7.1\,mV$.

## Dye adsorption
Equal volume of particle suspension (concentration, $C_{HL-HBPs}$) and dye solution (concentration, $C_0$) were mixed and maintained at room temperature for a preset time interval ($t$). Then the mixed solution was filtrated with a filter paper (pore size, $2\,\mu m$). Subsequently, the obtained solution was collected for UV–vis absorption measurement on a UV–vis spectrophotometer (Shimadzu, UV-2600), and further to calculate dye concentration ($C_t$), dye adsorption efficiency, and dye adsorption amount. The type and concentration of HL-HBPs, type of dyes, and t were verified for comparison. The dye adsorption efficiency and dye adsorption amount can be calculated as follows:

$$\text{Dye adsorption efficiency(\%)} = \frac{C_0 - 2C_t}{C_0} \times 100\% \tag{11}$$

$$\text{Dye adsorption amount} = \frac{C_0 - 2C_t}{C_{HL-HBPs}}(\mu g\,mg^{-1}) \tag{12}$$

## Dye recycle and wastewater treatment
Typically, the RB aqueous solution (20 ppm) was mixed with the PSS-PSDVB HL-HBPs aqueous suspension ($10\,mg\,mL^{-1}$) to achieve dye adsorption. Subsequently, the aqueous solution was separated by a filter paper (pore size, $2\,\mu m$), obtaining dyed particles and clean water. Then the PSS-PSDVB HL-HBPs adsorbed with RB were further dispersed in a mixture of ethanol and $n$-hexane (1:1 v/v) (or mixture of DCM and $n$-pentane (1:1 v/v)) to achieve dye desorption. After filtration by the filter paper, RB containing organic solution was obtained, leaving white particle powders. Finally, the organic solution was distilled at 80 °C (40 °C for the mixture of DCM and $n$-pentane (1:1 v/v)) to recover RB, and the particle powders can be used for repeated dye adsorption and desorption. The dye adsorption efficiency in water and desorption efficiency in organic solvent can be calculated according to the UV-vis absorption measurement.

## Dye adsorption kinetics characterized by in situ LSCM
Equal volume of particle suspension (PSS-PSDVB, $1.33\,mg\,mL^{-1}$) and RB solution (20 ppm) were mixed quickly at room temperature. Then $10\,\mu L$ of the mixed solution was encapsulated between two cover glass slides for LSCM visualization with a LSCM instrument affiliated with an optical microscope (Nikon, Eclipse Ti). LSCM images were acquired in situ at a preset time interval ($t$). Intensity profile measurement was used to obtain the fluorescent intensity of the HL-HBPs adsorbed with dyes.

## Data availability
The data supporting the findings of this study are available within the Article and its Supplementary Information. Other raw data generated during this study are available from the corresponding authors upon request. Source data are provided with this paper.

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

## Acknowledgements

S.W. acknowledges the support from the National Natural Science Foundation of China (22035008) and the National Key R&D Program of China (2019YFA0709300). Y.S. acknowledges the support from the National Key R&D Program of China (2022YFA1206900). Dengli Qiu and Yang Liu from Bruker, and Jingyun Fang from the Technical Institute of Physics and Chemistry, Chinese Academy of Sciences, are acknowledged for their help of AFM measurement. Ming Ji from Shanghai nateng Instruments Co., Ltd. is acknowledged for PiFM analysis. Ying Li and Xiang Li from Tsinghua University are acknowledged for TEM sample preparation and characterization. Jing Li from the Technical Institute of Physics and Chemistry, Chinese Academy of Sciences, is acknowledged for LSCM measurement. Xubo Liu from Technical Institute of Physics and Chemistry, Chinese Academy of Sciences, and Ao Hai from Beihang University, are acknowledged for time-lapse contact angle measurement. Zixin An from the Technical Institute of Physics and Chemistry, Chinese Academy of Sciences, is acknowledged for nuclear magnetic resonance (NMR) spectroscopy measurement. Hongyan Xiao from the Technical Institute of Physics and Chemistry, Chinese Academy of Sciences, is acknowledged for theoretical calculation discussion.

## Author contributions

S.W. proposed the research direction and guided the project. Y.S. performed most of the experiments and analyzed the data. J.Z. carried out the theoretical analysis. Z.Z. contributed to the AFM characterization and data analysis. Xiaoxia Li performed the dye separation and water treatment experiments. Y.Z. and X.S. synthesized the HL-HBPs. P.O'R. contributed to the PiFM characterization. Y.S. and J.Z. wrote the manuscript. Xiuling Li, X. Liang, L.J., and S.W. analyzed the data and revised the manuscript. All authors discussed the results and commented on the manuscript at all stages.

## Competing interests

The authors declare no competing interests.
