## [Peer Review File · Nature Communications]

Heterostructure particles enable omnidispersible in water and oil towards organic dye recycleReviewers' Comments:

Reviewer #1:

Remarks to the Author:

This study demonstrates the omnidispersity of hydrophilic-hydrophobic heterostructured particles (HL-HBPs), which are synthesized using a surface heterogeneous nano-structuring strategy. The as-synthesized HBPs were then applied for the adsorption and recycling of organic dyes. There is no doubt that the authors are experienced in synthesizing materials containing both hydrophobic and hydrophilic domains and have published excellent work previously. Therefore, the application needed to stand out with exceptional performance. Firstly, the dispersibility of HBPs should be backed up using theoretical models to explain the interaction and their ability to not aggregate in aqueous media. Secondly, the reasoning for achieved performance is not sound and the mechanism of adsorption requires to be supported with experimental/theoretical results, especially the fast equilibrium. Thirdly, given the low surface area, HBPs may not be suitable for dye adsorption, because their adsorption capacity of less than 15 mg/g is many folds lower than the materials reported in the literature. Finally, the claim that HBPs show promise for sustainable organic dye separation and recycling in an energy-saving and chemical-saving way is not substantiated, especially when membrane filtration (twice) and distillation steps are required for dye recycling. Therefore, this study is not recommended for publication.

Major Comments:

1. The premise of this work is sound and shows good promise. Well-dispersed colloidal particles could be applied in various fields as building blocks for self-assembly, as catalysts for energy conversion, and as therapeutic agents for tumor management. The buildup is interesting, especially the suitability of the surface heterogeneous nano-structuring strategy over the conventional surface modification strategy. However, the claimed application of HL-HBPs for sustainable water purification and organic dye recycling is not innovative or novel, particularly when adsorbents with exceptional dye separation and recycling have been reported in the literature. Additionally, it is not recommended to provide speculative statements without any comparison of the results achieved in this study vs. the literature.
2. In this study, HL-HBPs were synthesized by an emulsion interfacial polymerization approach, which has been employed by authors previously for the synthesis of materials with both hydrophobic and hydrophilic domains and have shown good performance for protein separation, glycopeptide separation, and bacterial separation. In the presence of these studies by authors and with the only major difference in the application (i.e., dye separation), this work does not raise to the level expected from work for publication in Nature Communication.
3. It is well-documented in literature that the surface area and pore size distribution can influence performance, particularly in the adsorption process. Firstly, there is a need to show the N₂ adsorption isotherms and discuss the basis for the selection of a model for pore size distribution. The onus is on the authors to discuss the implications of the results on the intended application (i.e., dye separation in this study). The authors are suggested to provide a time-lapse contact angle measurement using both water and organic solvents, which is also a good indicator of a material's hydrophilic and/or hydrophobic nature.
4. HL-HBPs with varied BET surface areas are synthesized (Figure S4). The overall surface area of HL-HBPs remains less than 15 m²/g, which should not be considered suitable for application in an adsorption process. Additionally, the adsorption/desorption curves of different HL-HBPs should be provided, and discuss the reasoning/hypothesis- why the authors believe that HL-HBPs could be a good adsorbent.
5. Yes, there is enough proof shown here that the HL-HBPs in this study have hydrophilic-hydrophobic surface heterogeneity, which seems to be overused to explain everything. However, at some stage, the authors are suggested to provide additional proof. For instance, It is claimed that 'positive or negative charges on hydrophilic domains offer interparticle repulsive force, preventing the particles from aggregating.' This could be supported by calculating the interaction potential of particles in aqueous media using DLVO theory which is simple yet describes particle interactions by combining electrostatic potential due to repulsive electrostatic double layer, and other attractive interactions.
6. The section on 'dye adsorption performance' indicates that the authors are not well-versed with the

latest trends in water treatment including dye wastewater treatment. Firstly, it is an oversimplification that water pollution is a direct reason for water scarcity. Secondly, obviously, Dye is discharged in the form of wastewater, so the research would obviously be focused on dye removal from water. Thirdly, the sludge issue in biological treatment is not related to dye sludge but biomass sludge, and it is to be noted that biological treatment should not be referred to as 'unsustainable' and 'not eco-friendly'. Fourthly, there are studies that focus on the recycling of the dyes and, with little effort, the authors will find suitable literature on this. Finally, all listed materials are not nano-porous (Line 166) and could just be called porous materials.

7. Indeed, it is difficult to explain the difference in the extent of dye rejection merely by electrostatic interaction. The authors may consider using some theoretical models (say DFT calculations) to calculate the adsorption energies of different dyes on HBPs or simply some other technique. The selectivity of an adsorbent is both good and bad, depending on the intended application.

8. The authors are suggested that the comparison of the results obtained in this study vs. literature should be based on identical experimental conditions. The authors did the comparison based on time, which understandably qualifies as one of the criteria of comparison. However, other experimental conditions such as adsorbent dose and dye concentration should be identical in all the studies. The HBP concentration of 10 mg/mL is on the higher side, while the dye concentration of 20 ppm is on the lower side as compared to some studies in the literature. It is an accepted norm that the comparison of adsorbents is made based on the adsorption capacity (mg/g). Based on the results of this study, unsurprisingly due to the poor surface area of HBPs, the maximum adsorption capacity (11.5 mg/g for MB and 13.62 mg/g for MG) is not very exciting and does not compare favorably with the state-of-the-art adsorbents such as LDH, Nanoparticles and/or commercial activated carbons. Therefore, the following claim: "we show the omnidispersible HL-HBPs enable an unprecedented simultaneous recycle of organic dyes and regeneration of HL-HBPs from the mimicked wastewater merely through the solvent exchange, surpassing most existing dye separation materials" does not hold.

9. In regard to equilibrium, the claim of fast equilibrium needs to be assessed by investigating the performance of HBPs at different concentrations. In addition, for universal applicability, performance over a wide of pH should be displayed and discussed. It is not clear the pH of the adsorption process in this study.

10. It is claimed several times that HBPs show promise for sustainable organic dye separation and recycling in an energy-saving and chemical-saving way. The authors should avoid such statements without proper economic analysis. In this study, according to the authors, membrane-based separation is utilized twice – firstly for the separation of HBPs from water, and secondly the separation of HBPs and organic dye. Finally, the distillation of organic solvent+organic dye for organic dye recovery. Could you please provide the characteristics of the membrane used in the 2nd step (i.e., separation of solvent+dye and HBPs). What were the total recovery of HBPs and organic dye because these materials are sticky and could cause severe membrane fouling? In all these steps, both energy and chemicals are required. Hence, the authors should not claim something that is not proven or demonstrated.

11. How about the changes in the properties of HBPs after 10 rounds and the purity of the organic dye? A simple NMR before and after the dye adsorption/desorption process should be able to show the purity levels.

Minor/Editing Comments:

1. It is suggested to use the term 'synthetic wastewater' instead of 'mimicked wastewater' throughout this manuscript and figure(s).
2. Line 155: what is ecology pollution? Do you want to say water pollution?
3. Line 157: "To treat the vast organic dye wastewater" – revise this sentence and make it simple.
4. Line 162-164: remove the word 'mediated'
5. Line 168: Change 'Common-use' to 'Commonly-used'
6. Line 232: change 'second' to 's'
7. suggestion to change 'heterostructured' to 'heterostructure'

Reviewer #2:

Remarks to the Author:

In this study, the unprecedented omnidispersity of hydrophilic-hydrophobic heterostructured particles (HL-HBPs) was synthesized. A surface heterogeneous nanostructuring strategy was used. Adhesion force images and Photo-induced force microscopy (PiFM) indicate the heterogeneous distributions on the surface. Organic dyes can be adsorbed from water and release them into organic solvents. Several questions need to be clarified.

1 More discussions about the advantages and disadvantages of surface heterogeneous nanostructuring strategy should be added.

2 The novelty of this study should be added in the abstract

3 More background of separation the organic dyes should be added.

4 Page 5 line 99 and (and 1,451 cm^{-1}), and 1,735 cm^{-1} should be checked

5 Procedures for Photo-induced force infrared spectra need to be added

6 Some abbreviations should be listed in tables, such as PSS31, PSDVB . PiFM and so on

7 Page 17 line 330 The dye adsorption efficiency slowly decreases in the several initial rounds, reasons should be given

8 Page 13 line 252 The kinetic analysis (figures) for the removal of dye in the adsorption process should be given. The comparison of similar materials or conventional materials need to be added

9 The adsorption mechanism should be added.

Reviewer #3:

Remarks to the Author:

In this work, the authors present a general strategy for heterogeneous surface nanostructuring for the preparation of particles with omnidispersibility in various solvents that can be used for recycling organic dyes. The strategy is based on the rational design of particles containing both hydrophilic and hydrophobic segments resulting from the heterogeneous emulsion polymerization process. As a result, the obtained HL-HPBs exhibit unique dispersion behavior in organic solvents and aqueous solutions, which is crucial for the rapid adsorption and desorption of organic dyes and the regeneration of the particles. Overall, the results of this work are well presented from both fundamental and technological points of view. The paper is well written. I would recommend the publication of this work in Nature Communications if the following points are addressed:

-The authors need to discuss in more detail why the adsorption/desorption efficiency of dyes decreases with increasing number of cycles. 84% adsorption efficiency after the 10th round doesn't seem that great, how can the recycling efficiency be further improved by the chemical design of the particles?

-The authors should provide a general table to compare the adsorption/desorption performance of dyes using HL-HPBs with other state-of-art materials in the literature in SI.

-For the adsorption index, it is not entirely clear if the interaction sites (electrostatic, hydrogen bonding, hydrophobic) contribute equally to this calculation? Basically, I would expect them to have different binding abilities with the charged/functional groups in the particles. Further discussion would be needed to distinguish the role of the different interaction sites.

Point-by-point Response

We greatly appreciate the reviewers' important comments and helpful suggestions that significantly improved the quality of our revised manuscript.

Response to Reviewer #1:

Reviewer #1 (Remarks to the Author):

This study demonstrates the omnidispersity of hydrophilic-hydrophobic heterostructured particles (HL-HBPs), which are synthesized using a surface heterogeneous nano-structuring strategy. The as-synthesized HBPs were then applied for the adsorption and recycling of organic dyes. There is no doubt that the authors are experienced in synthesizing materials containing both hydrophobic and hydrophilic domains and have published excellent work previously. Therefore, the application needed to stand out with exceptional performance. Firstly, the dispersibility of HBPs should be backed up using theoretical models to explain the interaction and their ability to not aggregate in aqueous media. Secondly, the reasoning for achieved performance is not sound and the mechanism of adsorption requires to be supported with experimental/theoretical results, especially the fast equilibrium. Thirdly, given the low surface area, HBPs may not be suitable for dye adsorption, because their adsorption capacity of less than 15 mg/g is many folds lower than the materials reported in the literature. Finally, the claim that HBPs show promise for sustainable organic dye separation and recycling in an energy-saving and chemical-saving way is not substantiated, especially when membrane filtration (twice) and distillation steps are required for dye recycling. Therefore, this study is not recommended for publication.

Response: We greatly appreciate the reviewer's important comments that help us to understand our work deeply. These helpful suggestions have significantly improved the quality of our revised manuscript. We are grateful that the reviewer recognized our previously published works on synthesizing materials containing both hydrophobic and hydrophilic domains. Those works have explored applications for biological separation,

from glycopeptide separation (*Adv. Mater.* **30**, 1803299 (2018).) to protein separation (*Adv. Mater.* **31**, 1900391 (2019); *Small* **17**, 2102802 (2022).) and bacterial separation (*Sci. Adv.* **3**, e1603203 (2017).). The novelty of present work is that unique omnidispersity of the synthesized particles can be achieved by the surface heterogeneous nanostructuring strategy, which was not revealed in our previous works. Furthermore, the unique omnidispersity of HL-HBPs enables maximal contact with, rapid adsorption and recovery of organic dyes in synthetic wastewater. Therefore, both omnidispersity and recyclability for organic dyes indicate their exceptional performance, and represent significant advances compared with our previous works. To address the reviewer's comments, we have carefully revised the manuscript after performing many additional experiments and theoretical calculations, and made comparisons to existing materials reported in the literatures. We believe the revised manuscript is suitable for publication in *Nature Communications*.

Firstly, according to the reviewer's suggestion, we have used Derjaguin-Landau-Verwey-Overbeek (DLVO) theory to calculate the interactions between HL-HBPs and to explain their ability to disperse in different solvents. The calculated total potential values in aqueous medium and oily medium both indicate a repulsive force between HL-HBPs to maintain dispersible. Calculation details are illustrated in the Response to Comment 5.

Secondly, it is a difficult task to clearly and quantitatively explain the adsorption mechanism based on the current adsorption theories. We have tried our best to understand what is behind this unique performance from the viewpoint of intense local electrostatic interactions. For fast equilibrium, we attributed the fast dye adsorption performance to the unique heterostructure of HL-HBPs. The outer hydrophilic domains are composed of negatively charged PSS with thickness of tens of nanometers, which provides intense electrostatic field for strong electrostatic interaction. Positively charged organic dyes in a long range ($\sim 1 \mu\text{m}$) can be attracted immediately to neutralize the charge of hydrophilic domains. Upon attracted towards the HL-HBPs, the hydrophobic domains on HL-HBPs provide short-range hydrophobic/ π - π bonding

interaction. HL-HBPs allow rapid diffusion and adsorption of organic dye molecules due to their wide pore size distribution from 10 nanometers to 100 nanometers (Supplementary Fig. 4c,d), which surpasses the microporous materials including MOFs and activated carbons (Supplementary Table 4). When the surface charge of HL-HBPs is neutralized, the diffusion of organic dyes towards the particles is almost halted, and adsorption equilibrium is achieved.

Thirdly, the surface area of our HL-HBPs is not high compared with most existing materials reported in the literatures. A large surface area certainly indicates a high adsorption capacity. However, a high surface area always suggests a small pore size, which is not beneficial for rapid diffusion and adsorption. For example, MOFs are microporous adsorbents exhibiting high surface area, but the adsorption equilibrium is always achieved in tens of hours. In our manuscript, we would not like to highlight the adsorption amount, but we want to show our HL-HBPs exhibit advantages in rapid adsorption. Dye adsorption equilibrium can be achieved in 5 s. Details are illustrated in the Response to Comments 1, 3, 4, and 8.

Fourthly, to avoid misleading the readers, we have revised our statements about sustainable organic dye separation. Some energy and chemicals are certainly required in all the steps, so we have deleted the statements, such as “sustainable organic dye separation”, “energy-saving”, and “chemical-saving” in the revised manuscript. Details are illustrated in the Response to Comment 10.

Major Comments:

1. The premise of this work is sound and shows good promise. Well-dispersed colloidal particles could be applied in various fields as building blocks for self-assembly, as catalysts for energy conversion, and as therapeutic agents for tumor management. The buildup is interesting, especially the suitability of the surface heterogeneous nanostructuring strategy over the conventional surface modification strategy. However, the claimed application of HL-HBPs for sustainable water purification and organic dye recycling is not innovative or novel, particularly when adsorbents with exceptional dye

separation and recycling have been reported in the literature. Additionally, it is not recommended to provide speculative statements without any comparison of the results achieved in this study vs. the literature.

Response: We greatly appreciate the reviewer's positive comments on the dispersity of particles and helpful suggestions on dye recycle. After comparison with adsorbent materials reported in the literature according to the reviewer's suggestions, the novelty of our particles is made more clearly, including rapid dye adsorption (Supplementary Table 4) and simultaneous regeneration of adsorbent materials and recycle of organic dyes (Supplementary Table 6). Firstly, the HL-HBPs demonstrate unique advantages in rapid adsorption, as dye adsorption equilibrium can be achieved in several seconds. In contrast, the adsorption equilibrium time of most existing adsorbent materials ranges from 5 min to tens of hours. Secondly, we propose an alternative and facile approach to desorb organic dyes from adsorbent materials. Conventional approaches mostly desorb dyes to regenerate adsorbent materials. In comparison, our strategy can simultaneously realize regeneration of adsorbent materials and recycle of organic dyes. To desorb organic dyes from materials reported in literatures, eluents containing inorganic acid, alkaline, or salt, are often added into the aqueous solution to weaken the interactions between materials and dyes. Such eluents make the dye recycle more complicated and require multiple steps. In comparison, organic solvent is used for dye desorption in our approach, and dyes can be recycled under a simple distillation process. Thus, adsorbent materials and organic dyes can be simultaneously recycled. Supplementary Tables 4 and 6 were added in the Supplementary Information to compare the difference. In addition, we added some perspectives discussing the future challenges and developments in the part of Conclusions.

Page 15, Line 283-287 in revised manuscript:

“The rapid dye adsorption equilibrium of HL-HBPs indicates their outstanding capability of dye adsorption, which is difficult to achieve with existing adsorbent materials, such as LDHs (from 5 minutes to tens of minutes)⁵⁸⁻⁶⁰, activated carbons (at least tens of minutes)⁵³⁻⁵⁵, and MOFs (tens of hours)^{51, 57} (Supplementary Table 4).”

Page 21, Line 405-411 in revised manuscript:

“The HL-HBPs provide a promising candidate for organic dye recycle compared with existing materials. To desorb organic dyes from materials reported in literatures, eluents containing inorganic acid, alkaline, or salt, are often added into the aqueous solution to weaken the interactions between materials and dyes (Supplementary Table 6). Such eluents make the dye recycle more complicated. In this study, organic solvent is used for dye desorption, and dyes can be recycled under a simple distillation process.”

Page 22, Line 422-425 in revised manuscript:

“Although the dye adsorption amount of HL-HBPs is lower than those existing materials with high surface area like MOFs, their adsorption rate is rapid. The next challenge is to create adsorbent materials with high adsorption capacity while maintaining the rapid adsorption performance.”

Supplementary Table 4:

Supplementary Table 4. Comparison of the pore size, BET surface area, dye adsorption capacity, and dye adsorption equilibrium time/kinetics of HL-HBPs with materials reported in literatures.

Materials	Pore size	BET surface area	Adsorption capacity	Adsorption equilibrium time/kinetics	Ref.
MOFs ([In ₃ O(COO) ₆] ⁺ -based)	0.28 nm~1.37 nm	1078 m ² g ⁻¹	20:1 in molar ratio (MOF:dye)	16 h~64 h	7
MOFs (NH ₂ -UiO-66)	~1 nm	1035 m ² g ⁻¹	Up to 697.7 mg g ⁻¹	Langmuir K _L : 0.006 g mg ⁻¹ min ⁻¹	8
Activated carbons (from woods)	/	/	~10 mg g ⁻¹	45 min	9
Activated carbons (Calgon, USA)	1.078 nm~1.088 nm	972 m ² g ⁻¹ 1~1015 m ² g ⁻¹	Up to 1.4 mmol g ⁻¹	> 700 h	10

Ferromagnetic hierarchical porous carbon	Wide range	260 m ² g ⁻¹	0.16 mg m ⁻²	~2 h	11
Carbon particles	1~10 nm	5.2 m ² g ⁻¹	Up to 79.5 mg g ⁻¹	10 s	12
Covalent organic polymers	0.51 nm, 0.76 nm, 1.36 nm	479 m ² g ⁻¹	/	30 min	13
LDHs (MgAl-LDH)	/	/	Up to 186 mg g ⁻¹	5 min	14
LDHs (NiAl-LDH)	3.4 nm	97 m ² g ⁻¹	150 mg g ⁻¹	6 min	15
Carbon-doped boron nitride	/	18.7 m ² g ⁻¹	Up to 747.10 mg g ⁻¹	2 h~3 h	16
MgO	/	154.85 m ² g ⁻¹	Up to 549.45 mg g ⁻¹	> 20 min	17
Chitosan hydrogel	4.34 nm~7.10 nm	2.15 m ² g ⁻¹ ~42.67 m ² g ⁻¹	Up to 1836 mg g ⁻¹	> 300 min	18
Amino grafted MCM-41	~3.4 nm	/	Up to 300 mg g ⁻¹	> 5 min	19
HL-HBPs	1 nm~100 nm	7.6 m ² g ⁻¹ ~14.8 m ² g ⁻¹	Up to 13.62 mg g ⁻¹	5 s	This work

Supplementary Table 6:

Supplementary Table 6. Comparison of the dye recycle performance for HL-HBPs and materials in literatures.

To desorb organic dyes from materials reported in literatures, eluents containing inorganic acid, alkaline, or salt, are often added into the aqueous solution to weaken the interactions between materials and dyes. Such eluents make the dye recycle more complicated. In comparison, organic solvent is used for dye desorption, and dyes can be recycled under a simple distillation process.

Materials	Recyclability of materials	Recyclability of dyes	Eluents	Ref.
MOFs	Yes	Not available	Salt (NaNO ₃)	7, 8
LDHs	Yes	Not available	Salt containing Cl ⁻ , NO ₃ ⁻ , or CO ₃ ²⁻	14, 15
Carbon-doped boron nitride	Yes	Not available	Acid (HCl)	16
MgO	Yes	Yes	Acid (HCl)	17
Chitosan hydrogel	Yes	Not available	Alkali (NaOH)	18
Amino grafted MCM-41	Yes	Not available	Alkali (NaOH)	19
HL-HBPs	Yes	Yes	Organic solvent	This work

References in Supplementary Information:

- Zhao, X. et al. Selective anion exchange with nanogated isoreticular positive metal-organic frameworks. *Nat. Commun.* **4**, 2344 (2013).
- Wang, H. et al. Membrane adsorbers with ultrahigh metal-organic framework loading for high flux separations. *Nat. Commun.* **10**, 4204 (2019).
- Heibati, B. et al. Kinetics and thermodynamics of enhanced adsorption of the dye AR 18 using activated carbons prepared from walnut and poplar woods. *J. Mol. Liq.* **208**, 99-105 (2015).
- Wang, S. & Zhu, Z. Effects of acidic treatment of activated carbons on dye adsorption. *Dyes Pigments* **75**, 306-314 (2007).
- Wang, D.-W., Li, F., Lu, G.Q. & Cheng, H.-M. Synthesis and dye separation performance of ferromagnetic hierarchical porous carbon. *Carbon* **46**, 1593-1599 (2008).
- Seifikar, F., Azizian, S. & Sillanpaa, M. Microwave-assisted synthesis of carbon powder for rapid dye removal. *Mater. Chem. Phys.* **250**, 123057 (2020).

13. Byun, J., Patel, H.A., Thirion, D. & Yavuz, C.T. Charge-specific size-dependent separation of water-soluble organic molecules by fluorinated nanoporous networks. *Nat. Commun.* **7**, 13377 (2016).
14. Sansuk, S., Srijaranai, S. & Srijaranai, S. A new approach for removing anionic organic dyes from wastewater based on electrostatically driven assembly. *Environ. Sci. Technol.* **50**, 6477-6484 (2016).
15. Pahalagedara, M.N. et al. Removal of azo dyes: Intercalation into sonochemically synthesized NiAl layered double hydroxide. *J. Phys. Chem. C* **118**, 17801-17809 (2014).
16. Wang, P., Wang, P., Guo, Y., Rao, L. & Yan, C. Selective recovery of protonated dyes from dye wastewater by pH-responsive BCN material. *Chem. Eng. J.* **412**, 128532 (2021).
17. Cao, N. et al. Superior selective adsorption of MgO with abundant oxygen vacancies to removal and recycle reactive dyes. *Sep. Purif. Technol.* **275** (2021).
18. Liu, Y. et al. Efficient removal and recycle of acid blue 93 dye from aqueous solution by acrolein crosslinked chitosan hydrogel. *Colloids Surf. Physicochem. Eng. Aspects* **632** (2022).
19. Rizzi, V. et al. Amino grafted MCM-41 as highly efficient and reversible ecofriendly adsorbent material for the Direct Blue removal from wastewater. *J. Mol. Liq.* **273**, 435-446 (2019).

2. In this study, HL-HBPs were synthesized by an emulsion interfacial polymerization approach, which has been employed by authors previously for the synthesis of materials with both hydrophobic and hydrophilic domains and have shown good performance for protein separation, glycopeptide separation, and bacterial separation. In the presence of these studies by authors and with the only major difference in the application (i.e., dye separation), this work does not raise to the level expected from work for publication in Nature Communication.

Response: We appreciate the reviewer's positive comments on our previous works. The emulsion interfacial polymerization approach is a general approach to synthesize heterostructure particles, which was proposed by our group in 2017 (*Sci. Adv.* **3**, e1603203 (2017)). The novelty of present work is that unique omnidispersity of the

synthesized particles can be achieved by the surface heterogeneous nanostructuring strategy, which was not revealed in our previous works. Furthermore, the unique omnidispersity of HL-HBPs enables maximal contact with, rapid adsorption and recovery of organic dyes in synthetic wastewater. Therefore, both omnidispersity and recyclability for organic dyes indicate their exceptional performance, and represent significant advances compared with our previous works.

3. It is well-documented in literature that the surface area and pore size distribution can influence performance, particularly in the adsorption process. Firstly, there is a need to show the N₂ adsorption isotherms and discuss the basis for the selection of a model for pore size distribution. The onus is on the authors to discuss the implications of the results on the intended application (i.e., dye separation in this study). The authors are suggested to provide a time-lapse contact angle measurement using both water and organic solvents, which is also a good indicator of a material's hydrophilic and/or hydrophobic nature.

Response: We greatly appreciate the reviewer's suggestions. First, we measured the surface area, pore size distribution, and N₂ adsorption-desorption isotherms of the HL-HBPs with various BET surface area (Supplementary Fig. 4c-d). Four HL-HBPs, with BET surface area of $7.6 \pm 0.1 \text{ m}^2 \text{ g}^{-1}$, $10.2 \pm 0.1 \text{ m}^2 \text{ g}^{-1}$, $13.0 \pm 0.1 \text{ m}^2 \text{ g}^{-1}$, and $14.8 \pm 0.1 \text{ m}^2 \text{ g}^{-1}$ (Mean \pm SD, n = 3), were prepared by regulating the addition volume of porogen (V_{Toluene}) in oil phase (Fig. 5d and Supplementary Fig. 4). Nitrogen adsorption-desorption isotherms indicate a type IV adsorption model. Pore size distribution graphs show the HL-HBPs exhibit micropores, mesopores, and macropores. Most of the pores exhibit pore diameter from 10 nanometers to 100 nanometers, which allows rapid diffusion and adsorption of organic dye molecules. Second, we measured the time-lapse contact angle of HL-HBPs, HLPs, and HBPs to indicate their hydrophilic and/or hydrophobic nature (Supplementary Fig. 6). The HBPs show a relatively hydrophobic nature, with contact angle of $136.0 \pm 2.7^\circ$ at 120 s. The HLPs show a relatively hydrophilic nature, with contact angle of $66.5 \pm 0.4^\circ$ at 120 s. The HL-HBPs show a

relatively hydrophilic nature compared with HBPs, and relatively hydrophobic nature compared with HLPs, with contact angle of $122.6 \pm 1.6^\circ$ at 120 s. These results suggest that the HL-HBPs are more favorable for water wetting compared with HBPs, facilitating their dispersion in water. Oil contact angle measurements show that oil droplets spread rapidly for all particles, probably due to the low interfacial tension of the oil (octane). All these results indicate superior omnidispersity performance of HL-HBPs for various solvents.

Page 18, Line 349-351 in revised manuscript:

“Most of the pores exhibit pore diameter from 10 nanometers to 100 nanometers (Supplementary Fig. 4c,d), which allows rapid diffusion and adsorption of organic dye molecules.”

Supplementary Fig. 4c-d:

Supplementary Fig. 4. Synthesis and characterizations of HL-HBPs with varied BET surface area. **c**, Nitrogen adsorption-desorption isotherms, and **d**, Pore size distribution graphs of HL-HBPs with BET surface area of $7.6 \pm 0.1 \text{ m}^2 \text{ g}^{-1}$, $10.2 \pm 0.1 \text{ m}^2 \text{ g}^{-1}$, $13.0 \pm 0.1 \text{ m}^2 \text{ g}^{-1}$, and $14.8 \pm 0.1 \text{ m}^2 \text{ g}^{-1}$ (Mean \pm SD, $n = 3$). The adsorption-desorption isotherms indicate a type IV adsorption model. Pore size distribution graphs show the HL-HBPs exhibit micropores, mesopores, and macropores. Most of the pores exhibit pore diameter from 10 nanometers to 100 nanometers.

Page 9, Line 148-152 in revised manuscript:

“Time-lapse contact angle measurement also indicates that the HL-HBPs show improved hydrophilicity compared with HBPs, and oil droplets can spread rapidly on a substrate adhered with HL-HBPs (Supplementary Fig. 6). All these results indicate

superior omnidispersity performance of HL-HBPs for various solvents.”

Supplementary Fig. 6:

Supplementary Fig. 6. Time-lapse contact angle measurement of HL-HBPs, HLPs,

and HBPs. a, Time-lapse water contact angle measurement in air for different particles.

The HBPs show a relatively hydrophobic nature, with contact angle of $136.0 \pm 2.7^\circ$ at 120 s.

The HLPs show a relatively hydrophilic nature, with contact angle of $66.5 \pm 0.4^\circ$ at 120 s.

The HL-HBPs show a relatively hydrophilic nature compared with HBPs, and relatively hydrophobic nature compared with HLPs, with contact angle of $122.6 \pm 1.6^\circ$ at 120 s.

These results suggest that the HL-HBPs are more favorable for water wetting compared with HBPs, facilitating their dispersion in water. Mean \pm SD, n = 3.

b, Oil contact angle measurement in air for different particles.

The oil droplets spread rapidly for all particles, probably due to the low interfacial tension of the oil (octane).

Contact angle measurement process is described as follows. The particle powders were adhered to a tape on a glass slide. Powders that were not tightly adhered were blew away.

A drop (2 μ L) of water or oil (octane) was dropped on the particles. A surface analyzer (LSA 100, LAUDA Scientific, GmbH) was used to determine the contact angles.

For water contact angle in air, the data were automatically calculated and recorded by the software.

For oil contact angle in air, the oil droplets can spread rapidly with a contact angle of $\sim 0^\circ$.

4. HL-HBPs with varied BET surface areas are synthesized (Figure S4). The overall surface area of HL-HBPs remains less than $15 \text{ m}^2/\text{g}$, which should not be considered suitable for application in an adsorption process. Additionally, the adsorption/desorption curves of different HL-HBPs should be provided, and discuss the reasoning/hypothesis why the authors believe that HL-HBPs could be a good adsorbent.

Response: Thanks for the reviewer’s suggestion. We measured the N_2 adsorption-desorption isotherms of the HL-HBPs with various BET surface area (Supplementary Fig. 4c). Nitrogen adsorption-desorption isotherms indicate a type IV adsorption model. Most of the pores exhibit pore diameter from 10 nanometers to 100 nanometers, which allows rapid diffusion and adsorption of organic dye molecules. Frankly speaking, the reviewer has proposed a very important suggestion. In our lab, we have been trying to overcome the great challenge of creating adsorbent materials with high adsorption capacity while maintaining the rapid adsorption performance. We have added N_2 adsorption-desorption isotherms of the HL-HBPs (Supplementary Fig. 4c) and some perspectives discussing the future challenges and developments in the part of Conclusions.

Page 18, Line 349-351 in revised manuscript:

“Most of the pores exhibit pore diameter from 10 nanometers to 100 nanometers (Supplementary Fig. 4c,d), which allows rapid diffusion and adsorption of organic dye molecules.”

Supplementary Fig. 4c:

Supplementary Fig. 4. Synthesis and characterizations of HL-HBPs with varied BET surface area. c, Nitrogen adsorption-desorption isotherms of HL-HBPs with BET surface area of $7.6 \pm 0.1 \text{ m}^2 \text{ g}^{-1}$, $10.2 \pm 0.1 \text{ m}^2 \text{ g}^{-1}$, $13.0 \pm 0.1 \text{ m}^2 \text{ g}^{-1}$, and $14.8 \pm 0.1 \text{ m}^2 \text{ g}^{-1}$ (Mean \pm SD, $n = 3$).

Page 22, Line 422-425 in revised manuscript:

“Although the dye adsorption amount of HL-HBPs is lower than those existing materials with high surface area like MOFs, their adsorption rate is rapid. The next challenge is to create adsorbent materials with high adsorption capacity while maintaining the rapid adsorption performance.”

5. Yes, there is enough proof shown here that the HL-HBPs in this study have hydrophilic-hydrophobic surface heterogeneity, which seems to be overused to explain everything. However, at some stage, the authors are suggested to provide additional proof. For instance, it is claimed that ‘positive or negative charges on hydrophilic domains offer interparticle repulsive force, preventing the particles from aggregating.’ This could be supported by calculating the interaction potential of particles in aqueous media using DLVO theory which is simple yet describes particle interactions by combining electrostatic potential due to repulsive electrostatic double layer, and other attractive interactions.

Response: Thanks for the reviewer’s constructive suggestion. To elucidate the unconventional dispersion property of the HL-HBPs, we calculated the interaction potential between two HL-HBPs using classical DLVO theory (Fig. 3d). Results show that the $V_{\text{total, w}}$ of HL-HBPs in water is positive at $d > 0$ and reaches maximum of $\sim 2,873 \text{ k}_B\text{T}$ for negatively charged PSS-PSDVB HL-HBPs ($\zeta = -45.8 \text{ mV}$), and $\sim 2,511 \text{ k}_B\text{T}$ for positively charged PTMAEMC-PSDVB HL-HBPs ($\zeta = +42.9 \text{ mV}$), suggesting the dispersion of HL-HBPs in water is quite stable (Fig. 3e). In octane, $V_{\text{vdW, o}}$ and $V_{\text{DL, o}}$ are much reduced compared with those in water, and $V_{\text{DL, o}}$ is not screened in a long range, resulting in a maximum $V_{\text{total, o}}$ of $\sim 14 \text{ k}_B\text{T}$ for negatively charged PSS-PSDVB HL-HBPs ($\zeta = -19.9 \text{ mV}$), and $\sim 2 \text{ k}_B\text{T}$ for positively charged PTMAEMC-PSDVB HL-

HBP ($\zeta = +7.1$ mV). These results indicate that the dispersion of HL-HBPs in oil is stable (Fig. 3f). Therefore, the unique HL-HBPs can be dispersed and maintain stable in both water and oil. Theoretical calculations were added in the revised manuscript.

Page 9, Line 153-Page 10, Line 181 in revised manuscript:

Theoretical calculations for the dispersion mechanism of the HL-HBPs.

To elucidate the unconventional dispersion property of the HL-HBPs, we calculated the interaction potential between two HL-HBPs using classical Derjaguin-Landau-Verwey-Overbeek (DLVO) theory (Fig. 3d-f). In typical cases of water and oil (octane), two HL-HBPs are separated by a distance d (Fig. 2d). The pair potential of HL-HBPs (V_{total}) is attributed to the sum of van der Waals interaction potential (V_{vdW}) and electrostatic interaction potential (V_{DL}), i.e., $V_{\text{total}} = V_{\text{vdW}} + V_{\text{DL}}$. Firstly, the V_{vdW} was calculated according to the Hamaker model³⁶, in which the HL-HBPs were treated as core-shell particles³⁷ (see Methods). It should be noted that the Hamaker constant of the HL-HBPs are also affected by the oil since oil can permeate into the pores of the particles (Supplementary Fig. 7). Secondly, the V_{DL} resulting from the charged hydrophilic domains was calculated by the Poisson-Boltzmann equation³⁸. Two HL-HBPs, including negatively charged PSS-PSDVB and positively charged poly(2-trimethylammoniumethyl methacrylate chloride)-PSDVB (PTMAEMC-PSDVB), were calculated for comparison. The results show that the $V_{\text{total, w}}$ of HL-HBPs in water is positive at $d > 0$ and reaches maximum of $\sim 2,873$ kBT for negatively charged PSS-PSDVB HL-HBPs ($\zeta = -45.8$ mV), and $\sim 2,511$ kBT for positively charged PTMAEMC-PSDVB HL-HBPs ($\zeta = +42.9$ mV) (Fig. 2e), suggesting the dispersion of HL-HBPs in water is quite stable. In octane, $V_{\text{vdW, o}}$ and $V_{\text{DL, o}}$ are much reduced compared with those in water, and $V_{\text{DL, o}}$ is not screened in a long range, resulting in a maximum $V_{\text{total, o}}$ of ~ 14 kBT for negatively charged PSS-PSDVB HL-HBPs ($\zeta = -19.9$ mV), and ~ 2 kBT for positively charged PTMAEMC-PSDVB HL-HBPs ($\zeta = +7.1$ mV) (Fig. 2f). These results indicate that the dispersion of HL-HBPs in oil is stable. The omnidispersity of HL-HBPs can be attributed to the unique hydrophilic-hydrophobic heterostructure. First, the hydrophilic domains are favorable for the wetting of water, and the

hydrophobic domains are favorable for the wetting of oil, making it easy for both water and oil to fill the gaps between particles. Second, introduction of charge on hydrophilic domains offers interparticle repulsive force, preventing the particles from aggregating. Therefore, the unique HL-HBPs are dispersible in both water and oil.

Fig. 3d-f:

Fig. 3 | Dispersion performance of the HL-HBPs. **d**, Scheme for interaction potential calculation. The V_{total} between two HL-HBPs separated by a distance d is calculated according to attractive V_{vdW} and repulsive V_{DL} . **e**, V_{vdW} , V_{DL} , and V_{total} between two HL-HBPs in water. **f**, V_{vdW} , V_{DL} , and V_{total} between two HL-HBPs in oil (octane). A positive V_{total} indicates repulsive force and stable dispersity. V_{vdW} : van der Waals interaction potential. V_{DL} : electrostatic interaction potential. V_{total} : total interaction potential.

Methods (Page 26, Line 501-Page 27, Line 535, in revised manuscript):

Theoretical calculations for the dispersion mechanism. *Calculations of V_{vdW} .* The V_{vdW} was calculated by the Hamaker model for the cases of water and octane, respectively. To simplify calculation, the HL-HBPs are treated as core-shell particles. Two HL-HBPs of radius R and shell thickness of δ are separated by a distance d . The Hamaker constants of the core, the shell and the medium solvent are A_c , A_s , and A_m , respectively. We assume the pores of HL-HBPs are filled with air when they are dispersed in water, while filled with oil when they are dispersed in oil (Supplementary Fig. 7), and the corresponding Hamaker constants are given by³⁷

$$A_c = \left((1-p)A_{PSDVB}^{1/2} + pA_m^{1/2} \right)^2$$

$$A_s = \left((1-p)A_{PSS}^{1/2} + pA_m^{1/2} \right)^2$$

where p represents the particle porosity, which is obtained from the pore volume when the particle density is assumed to be 1 g cm^{-3} . Here the Hamaker constant of pure substance 1 is calculated by⁸

$$A = \frac{3}{4} k_B T \left(\frac{\varepsilon_1 - \varepsilon_3}{\varepsilon_1 + \varepsilon_3} \right)^2 + \frac{3h\nu_e}{16\sqrt{2}} \left(\frac{(n_1^2 - n_3^2)^2}{(n_1^2 + n_3^2)^{3/2}} \right)$$

where $k_B = 1.38 \times 10^{-23} \text{ J K}^{-1}$, $T = 300 \text{ K}$, $h = 6.62 \times 10^{-34} \text{ J s}$, $\nu_e = 3 \times 10^{15} \text{ s}^{-1}$, $\varepsilon_3 = 1$, and $n_3 = 1$ for vacuum³⁷. The values used for calculation are shown in Supplementary Table 2.

The V_{vdW} of the HL-HBPs are given by³⁷

$$V_{vdW} = -\frac{1}{12} \left[H_{ss}(A_s^{1/2} - A_m^{1/2})^2 + H_{cc}(A_c^{1/2} - A_s^{1/2})^2 + 2H_{cs}(A_c^{1/2} - A_s^{1/2})(A_s^{1/2} - A_m^{1/2}) \right]$$

where the unretarded H functions are given by

$$H(x,y) = \frac{y}{x^2 + xy + x} + \frac{y}{x^2 + xy + x + y} + 2 \ln \left[\frac{x^2 + xy + x}{x^2 + xy + x + y} \right]$$

$$x = \frac{\Delta}{2r_1}, y = r_2/r_1$$

$$H_{ss}: \Delta = d, r_1 = r_2 = R$$

$$H_{cc}: \Delta = d + 2\delta, r_1 = r_2 = R - \delta$$

$$H_{cs}: \Delta = d + \delta, r_1 = R, r_2 = R - \delta$$

Calculations of V_{DL} . The V_{DL} is calculated by the Poisson-Boltzmann equation

$$V_{DL} = 2\pi\varepsilon_0\varepsilon_r R \zeta^2 \ln(1 + e^{-\kappa d})$$

where $\varepsilon_0 = 8.85 \times 10^{-12} \text{ Farad m}^{-1}$, ε_r is the dielectric constant of the solvent, ζ is the zeta potential of the particles in the solvent, κ^{-1} is the Debye length.

For HL-HBPs dispersed in water, we used $\varepsilon_r = 80.2$, $\kappa^{-1} = 100 \text{ nm}$. In addition, ζ (PSS-PSDVB) = -45.8 mV , ζ (PTMAEMC-PSDVB) = $+42.9 \text{ mV}$.

For HL-HBPs dispersed in octane, we used $\varepsilon_r = 1.948$. To simplify the calculation process, we assumed $\kappa^{-1} = 100,000 \text{ nm}$ since the Debye length dramatically increases owing to the weak ionic shielding effect in solvents with low dielectric constant⁶⁴. In

addition, ζ (PSS-PSDVB) = -19.9 mV, ζ (PTMAEMC-PSDVB) = +7.1 mV.

References:

8. Israelachvili, J.N. Intermolecular and surface forces. (Academic Press, 2011).
36. Hamaker, H.C. The London-van der Waals attraction between spherical particles. *Physica* **4**, 1058-1072 (1937).
37. Vincent, B. Vanderpol attraction between colloid particles having adsorbed layers. 2. Calculation of interaction curves. *J. Colloid Interf. Sci.* **42**, 270-285 (1973).
38. Goodwin, J.W. Colloids and interfaces with surfactants and polymers-An introduction. (John Wiley & Sons, 2004).
64. Briscoe, W.H. & Horn, R.G. Direct measurement of surface forces due to charging of solids immersed in a nonpolar liquid. *Langmuir* **18**, 3945-3956 (2002).

Supplementary Table 2:

Supplementary Table 2. Parameters of employed materials for calculation.

Material	Dielectric constant (ϵ_1)	Refractive index (n_1)	Hamaker constant (A, 10^{-20} J)
PSDVB	2.4 ¹	1.6 ²	9.601
PSS	2.56 ^{3,4}	1.5335 ^a	7.906
Water	80.2 ²	1.333 ²	3.733
Octane	1.948 ²	1.398 ¹	4.759
Air	1.000585	1.0003	3.380×10^{-6}

^aThe value was obtained in Supplementary Fig. 8.

Supplementary Fig. 7:

Supplementary Fig. 7. LSCM image of HL-HBPs dispersed in C6 labeled octane solution. The image implies that octane can permeate into the interior of HL-HBPs. The LSCM image was obtained on LSCM instruments affiliated to an optical microscope (Nikon, Eclipse Ti).

Supplementary Fig. 8:

Supplementary Fig. 8. Relationship between refractive index (n) and PSS mass fraction (w) in aqueous solution. The refractive index (n) of pure PSS was predicted by measuring the refractive indices of PSS aqueous solutions with different PSS mass fraction (w). A series of PSS (average $M_w = 70,000$, purchased from Sigma Aldrich) aqueous solutions were prepared with PSS mass fraction of 5%, 10%, 20%, 30%, and 40%, respectively. Subsequently, their refractive indices were measured with an Abbe refractometer at 20 °C ($n_{20/D}$) ($n = 3$). The relationship between n and w can be linearly fitted⁵ with an equation of $n = 0.00204 \times w + 1.3295$ ($R^2 = 0.99881$). When pure PSS

is applied ($w = 100\%$), $n = 1.5335$, which is in accordance with previous study⁶.

References in Supplementary Information:

1. Brandrup, J. Polymer handbook (4th edition). (John Wiley and Sons, 1999).
2. Speight, J.G. Lange's handbook of chemistry. (The McGraw-Hill Companies, Inc., 2004).
3. Zong, Y., Xu, F., Su, X. & Knoll, W. Quartz crystal microbalance with integrated surface plasmon grating coupler. *Anal. Chem.* **80**, 5246-5250 (2008).
4. Sheng Hsiung, C., Chien-Hung, C., Feng-Sheng, K., Chuen-Lin, T. & Chun-Guey, W. Unraveling the enhanced electrical conductivity of PEDOT:PSS thin films for ITO-free organic photovoltaics. *IEEE Photonics J.* **6**, 1-7 (2014).
5. Roots, J. & Nyström, B. Concentration and temperature dependence of the refractive index increment of a polystyrene sample in trans-decalin. *J. Polym. Sci. Polym. Phys. Ed.* **16**, 695-701 (1978).
6. Ono, Y., Nakase, I., Matsumoto, A. & Kojima, C. Rapid optical tissue clearing using poly(acrylamide-co-styrenesulfonate) hydrogels for three-dimensional imaging. *J. Biomed. Mater. Res. B Appl. Biomater.* **107**, 2297-2304 (2019).

6. The section on ‘dye adsorption performance’ indicates that the authors are not well-versed with the latest trends in water treatment including dye wastewater treatment. Firstly, it is an oversimplification that water pollution is a direct reason for water scarcity. Secondly, obviously, Dye is discharged in the form of wastewater, so the research would obviously be focused on dye removal from water. Thirdly, the sludge issue in biological treatment is not related to dye sludge but biomass sludge, and it is to be noted that biological treatment should not be referred to as ‘unsustainable’ and ‘not eco-friendly’. Fourthly, there are studies that focus on the recycling of the dyes and, with little effort, the authors will find suitable literature on this. Finally, all listed materials are not nano-porous (Line 166) and could just be called porous materials.

Response: Thanks for the reviewer’s suggestions. We have carefully revised related statements in the revised manuscript.

Firstly, according to the reviewer’s suggestion, we have revised the statement that “The global production of organic dyes approaches 700,000 tons annually, however,

nearly 10-15% of them is discharged into industrial and household wastewater, raising the risks of ecology pollution, water scarcity, and public health.” to “The global production of organic dyes approaches 700,000 tons annually, however, nearly 10-15% of them is discharged into industrial and household wastewater, which has become an important source of water pollution and a non-negligible threat for public health⁴¹⁻⁴⁴.” (Page 11, Line 191-194 in revised manuscript).

Secondly, we certainly agree with the reviewer that the research should be mainly focused on the dye removal from water. Nevertheless, in addition to address the water pollution issue caused by the organic dyes, we also attempt to recycle organic dyes from the wastewater. According to our results, the recovered dyes have rarely changed after the adsorption-desorption process, implying their successful recycle. Therefore, our research provides a green approach to achieve repeated use of organic dyes, which is beneficial for reducing organic dye pollution in printing and dyeing industry.

Thirdly, according to the reviewer’s suggestion, we have revised the statement that “Conventional methods, including coagulation-flocculation and biological degradation, can remove part of the organic dyes from wastewater, after which the sludge is always dumped. These methods are neither sustainable nor eco-friendly due to the generation of secondary wastes.” to “Conventional coagulation-flocculation method can remove part of the organic dyes from wastewater, after which the dye-containing sludge is always dumped^{45, 46}. In addition, this approach is mainly suitable for water insoluble dyes, but generally not suitable for water soluble dyes. Alternatively, biological degradation uses biological materials, such as algae, bacteria, fungi, and yeasts, to disintegrate organic dyes. This approach also generates biomass sludge⁴⁷.” (Page 11, Line 196-201 in revised manuscript).

Fourthly, we note that there are some literatures that report dye recycle. However, after careful evaluation, we find that most of them can achieve the recycle of adsorbent materials, and minority of them can achieve the recycle of organic dyes. We compared the dye recycle performance with the literature (Supplementary Table 6). To desorb organic dyes from materials reported in literatures, eluents containing inorganic acid,

alkaline, or salt, are often added into the aqueous solution to weaken the interactions between materials and dyes. Such eluents make the dye recycle more complicated. In our study, organic solvent is used for dye desorption, and dyes can be recycled under a simple distillation process. Supplementary Table 6 was added in the Supplementary Information to compare the difference.

Finally, we have revised “nanoporous” to “porous” (Page 11, Line 207 in revised manuscript).

Page 21, Line 405-411 in revised manuscript:

“The HL-HBPs provide a promising candidate for organic dye recycle compared with existing materials. To desorb organic dyes from materials reported in literatures, eluents containing inorganic acid, alkaline, or salt, are often added into the aqueous solution to weaken the interactions between materials and dyes (Supplementary Table 6). Such eluents make the dye recycle more complicated. In this study, organic solvent is used for dye desorption, and dyes can be recycled under a simple distillation process.”

Supplementary Table 6:

Supplementary Table 6. Comparison of the dye recycle performance for HL-HBPs and materials in literatures. To desorb organic dyes from materials reported in literatures, eluents containing inorganic acid, alkaline, or salt, are often added into the aqueous solution to weaken the interactions between materials and dyes. Such eluents make the dye recycle more complicated. In comparison, organic solvent is used for dye desorption, and dyes can be recycled under a simple distillation process.

Materials	Recyclability of materials	Recyclability of dyes	Eluents	Ref.
MOFs	Yes	Not available	Salt (NaNO ₃)	7, 8
LDHs	Yes	Not available	Salt containing Cl ⁻ , NO ₃ ³⁻ , or CO ₃ ²⁻	14, 15
Carbon-doped boron nitride	Yes	Not available	Acid (HCl)	16

MgO	Yes	Yes	Acid (HCl)	17
Chitosan hydrogel	Yes	Not available	Alkali (NaOH)	18
Amino grafted MCM-41	Yes	Not available	Alkali (NaOH)	19
HL-HBPs	Yes	Yes	Organic solvent	This work

References in Supplementary Information:

7. Zhao, X. et al. Selective anion exchange with nanogated isoreticular positive metal-organic frameworks. *Nat. Commun.* **4**, 2344 (2013).
8. Wang, H. et al. Membrane adsorbers with ultrahigh metal-organic framework loading for high flux separations. *Nat. Commun.* **10**, 4204 (2019).
14. Sansuk, S., Srijaranai, S. & Srijaranai, S. A new approach for removing anionic organic dyes from wastewater based on electrostatically driven assembly. *Environ. Sci. Technol.* **50**, 6477-6484 (2016).
15. Pahalagedara, M.N. et al. Removal of azo dyes: Intercalation into sonochemically synthesized NiAl layered double hydroxide. *J. Phys. Chem. C* **118**, 17801-17809 (2014).
16. Wang, P., Wang, P., Guo, Y., Rao, L. & Yan, C. Selective recovery of protonated dyes from dye wastewater by pH-responsive BCN material. *Chem. Eng. J.* **412**, 128532 (2021).
17. Cao, N. et al. Superior selective adsorption of MgO with abundant oxygen vacancies to removal and recycle reactive dyes. *Sep. Purif. Technol.* **275** (2021).
18. Liu, Y. et al. Efficient removal and recycle of acid blue 93 dye from aqueous solution by acrolein crosslinked chitosan hydrogel. *Colloids Surf. Physicochem. Eng. Aspects* **632** (2022).
19. Rizzi, V. et al. Amino grafted MCM-41 as highly efficient and reversible ecofriendly adsorbent material for the Direct Blue removal from wastewater. *J. Mol. Liq.* **273**, 435-446 (2019).

7. Indeed, it is difficult to explain the difference in the extent of dye rejection merely

by electrostatic interaction. The authors may consider using some theoretical models (say DFT calculations) to calculate the adsorption energies of different dyes on HBPs or simply some other technique. The selectivity of an adsorbent is both good and bad, depending on the intended application.

Response: Thanks for the reviewer's suggestion. We have consulted with Prof. Dr. Hongyan Xiao from Technical Institute of Physics and Chemistry, Chinese Academy of Sciences. She has rich experience in DFT calculations (quantum chemistry computation), including molecular dynamics study of interfaces, structure-property analysis of molecules, and catalytic reaction mechanism. Prof. Xiao has tried her best to calculate the adsorption energies of different dyes on HL-HBPs. Calculation results imply that there is no obvious law between the adsorption energies of different dyes and the adsorption amounts. After an in-depth discussion, the proposed adsorption index in the manuscript can be used as a simple model to roughly understand the adsorption difference.

Selected publications of Prof. Xiao are listed as follows.

1. Yang, L., Xiao, H., Qian, Y., Zhao, X., Kong, X.Y., Liu, P., Xin, W., Fu, L., Jiang, L. & Wen, L. Bioinspired hierarchical porous membrane for efficient uranium extraction from seawater. *Nat. Sustain.* **5**, 71-80 (2022).
2. Lei, T., Zhou, C., Huang, M.Y., Zhao, L.M., Yang, B., Ye, C., Xiao, H., Meng, Q.Y., Ramamurthy, V., Tung, C.H. & Wu, L.Z. General and efficient intermolecular [2+2] photodimerization of chalcones and cinnamic acid derivatives in solution through visible light catalysis. *Angew. Chem. Int. Ed.* **56**, 15407-15410 (2017).
3. Grubb, M.P., Warter, M.L., Xiao, H., Maeda, S., Morokuma, K. & North, S.W. No straight path: Roaming in both ground- and excited-state photolytic channels of $\text{NO}_3 \rightarrow \text{NO} + \text{O}_2$. *Science* **335**, 1075-1078 (2012).

8. The authors are suggested that the comparison of the results obtained in this study vs. literature should be based on identical experimental conditions. The authors did the comparison based on time, which understandably qualifies as one of the criteria of

comparison. However, other experimental conditions such as adsorbent dose and dye concentration should be identical in all the studies. The HBP concentration of 10 mg/mL is on the higher side, while the dye concentration of 20 ppm is on the lower side as compared to some studies in the literature. It is an accepted norm that the comparison of adsorbents is made based on the adsorption capacity (mg/g). Based on the results of this study, unsurprisingly due to the poor surface area of HBPs, the maximum adsorption capacity (11.5 mg/g for MB and 13.62 mg/g for MG) is not very exciting and does not compare favorably with the state-of-the-art adsorbents such as LDH, Nanoparticles and/or commercial activated carbons. Therefore, the following claim: “we show the omnidispersible HL-HBPs enable an unprecedented simultaneous recycle of organic dyes and regeneration of HL-HBPs from the mimicked wastewater merely through the solvent exchange, surpassing most existing dye separation materials” does not hold.

Response: Thanks for the reviewer’s suggestion. As the reviewer commented, the BET surface area of the HL-HBPs is not very high compared with existing adsorbent materials, and thus resulting in a relatively lower adsorption capacity. However, rapid dye adsorption of the HL-HBPs is the advantage we want to highlight in this work. In the future, we would like to further improve the adsorption capacity. According to the reviewer’s kind suggestions and comments, we corrected the statement about the advantages of HL-HBPs for rapid dye adsorption in revised manuscript. In addition, we added some perspectives discussing the future challenges and developments in the part of Conclusions.

Page 4, Line 77-79 in revised manuscript:

“we show the omnidispersible HL-HBPs enable an unprecedented simultaneous recycle of organic dyes and regeneration of HL-HBPs from the synthetic wastewater merely through solvent exchange.”

Page 22, Line 422-425 in revised manuscript:

“Although the dye adsorption amount of HL-HBPs is lower than those existing materials with high surface area like MOFs, their adsorption rate is rapid. The next

challenge is to create adsorbent materials with high adsorption capacity while maintaining the rapid adsorption performance.”

9. In regard to equilibrium, the claim of fast equilibrium needs to be assessed by investigating the performance of HBPs at different concentrations. In addition, for universal applicability, performance over a wide of pH should be displayed and discussed. It is not clear the pH of the adsorption process in this study.

Response: We greatly appreciate the reviewer’s kind suggestions. We tested the dye adsorption performance at different concentration of HL-HBPs (from 1.33 mg mL⁻¹ to 3.33 mg mL⁻¹, 6.67 mg mL⁻¹, 10 mg mL⁻¹, and 13.33 mg mL⁻¹), and different pH values (from 0 to 14). The adsorption equilibrium was almost achieved at adsorption time of 5 s, and adsorption efficiency rarely increased in 5 minutes (Fig. 5b in revised manuscript). In original manuscript of the study, the solution used for adsorption process was deionized water, and the pH was about 5.50. In revised manuscript, we also measured dye adsorption efficiency at different pH value. The dye adsorption efficiency maintains >92% at pH range of 1-14 without significant decrement. When pH = 0, the dye adsorption efficiency slightly decreases to 86.15 ± 1.09% (Mean ± SD, n = 3) (Supplementary Fig. 10). These results indicate the dye adsorption can be achieved in a wide range of pH value.

Page 15, Line 279-283 in revised manuscript:

“Quantificationally, the dye adsorption efficiency was tested for kinetics study when the concentration of HL-HBPs varied from 1.33 mg mL⁻¹ to 3.33 mg mL⁻¹, 6.67 mg mL⁻¹, 10 mg mL⁻¹, and 13.33 mg mL⁻¹. The adsorption equilibrium was almost achieved at adsorption time of 5 s, and adsorption efficiency rarely increased in 5 min (Fig. 5b).”

Page 17, Line 323-326 in revised manuscript:

“The dye adsorption efficiency maintains >92% at pH range of 1-14 without significant decrement, and slightly decreases to 86.15 ± 1.09% (Mean ± SD, n = 3) when pH = 0 (Supplementary Fig. 10), indicating the dye adsorption can be achieved in a wide range of pH value.”

Fig. 5b:

Fig. 5 | Dye adsorption kinetics and mechanism with the HL-HBPs. b, RB adsorption efficiency with PSS-PSDVB HL-HBPs of various particle concentration ($C_{HL-HBPs}$) at different adsorption time. The initial concentration of RB solution is 20 ppm. Dye adsorption efficiency, Mean \pm SD, n = 3.

Supplementary Fig. 10:

Supplementary Fig. 10. Adsorption of RB with HL-HBPs at pH range of 0-14. The dye adsorption efficiency maintains $>92\%$ at pH range of 1-14 without significant decrement. When pH = 0, the dye adsorption efficiency slightly decreases to $86.15 \pm 1.09\%$ (Mean \pm SD, n = 3). These results indicate the dye adsorption can be achieved in a wide range of pH value.

10. It is claimed several times that HBPs show promise for sustainable organic dye separation and recycling in an energy-saving and chemical-saving way. The authors should avoid such statements without proper economic analysis. In this study, according to the authors, membrane-based separation is utilized twice – firstly for the separation of HBPs from water, and secondly the separation of HBPs and organic dye. Finally, the distillation of organic solvent+organic dye for organic dye recovery. Could you please provide the characteristics of the membrane used in the 2nd step (i.e., separation of solvent+dye and HBPs). What were the total recovery of HBPs and organic dye because these materials are sticky and could cause severe membrane fouling? In all these steps, both energy and chemicals are required. Hence, the authors should not claim something that is not proven or demonstrated.

Response: Thanks for the reviewer’s kind suggestions. Some energy and chemicals are certainly required in all the steps, so we have deleted the statements, such as “sustainable organic dye separation”, “energy-saving”, and “chemical-saving” in the revised manuscript.

According to the reviewer’s suggestions, we also investigated the property changes of the membrane (Supplementary Fig. 14). The original membranes are composed of nylon (polyamide), with average pore diameter of 2 μm . Such pore size allows the fast flow of water and retention of particles. After 1st step filtration, the HL-HBPs adsorbed with dyes are retained on the membrane. Most of the HL-HBPs can be rinsed out using organic solvent, leaving minority of them retained on the membranes. After 2nd step filtration, the HL-HBPs are retained on the membrane. Similarly, most of the HL-HBPs can be rinsed out using water, leaving minority of them retained on the membranes. The recovery percentages of HL-HBPs and dyes are 95.5% and 92.2%, respectively. Therefore, membrane fouling exists but is not severe. The property changes of membranes after filtration were added as Supplementary Fig. 14 in the revised manuscript.

Page 21, Line 399-403 in revised manuscript:

“Minority of the HL-HBPs can be retained on the membrane after the adsorption-desorption process (Supplementary Fig. 14), and the regeneration ratio of the HL-HBPs is about 95.5%. After one round of adsorption-desorption process, the recycle ratio of the dyes is about 92.2%.”

Supplementary Fig. 14:

Supplementary Fig. 14. Photos and SEM images of original membranes and membranes after 1st step and 2nd step filtration. a, The original membranes are composed of nylon (polyamide), with average pore diameter of 2 μm. Such pore size allows the fast flow of water and retention of particles. **b,** The HL-HBPs adsorbed with dyes are retained on the membrane after 1st step filtration. Most HL-HBPs can be rinsed out using organic solvent, leaving minority of them retained on the membranes. **c,** The HL-HBPs are retained on the membrane after 2nd step filtration. Most HL-HBPs can be rinsed out using water, leaving minority of them retained on the membranes. The total recovery ratio of HL-HBPs is about 95.5%.

11. How about the changes in the properties of HBPs after 10 rounds and the purity of the organic dye? A simple NMR before and after the dye adsorption/desorption process should be able to show the purity levels.

Response: Thanks for the reviewer’s helpful suggestions. We used UV-vis absorption spectra to characterize the HL-HBPs before and after 10 rounds of dye adsorption-desorption. In addition, NMR spectra were used to characterize the dyes before and

after dye adsorption-desorption. UV-vis absorption spectra (Wave length: 300-700 nm) of the HL-HBPs have rarely changed after 10 rounds of adsorption-desorption process, including the characteristic absorption peak of the dyes ($\lambda_{\text{max, RB}} = 553.5 \text{ nm}$), indicating that dyes are rarely retained on the HL-HBPs (Supplementary Fig. 12). ^1H NMR spectra show that the chemical shift positions have rarely changed after the adsorption-desorption process, indicating that there are rare additional impurities in the recycled dyes (Supplementary Fig. 13).

Page 21, Line 397-399 in revised manuscript:

“The regenerated HL-HBPs have rarely changed after 10 rounds of adsorption-desorption process, and there are rare additional impurities in the recycled dyes (Supplementary Figs. 12-13).”

Supplementary Fig. 12:

Supplementary Fig. 12. UV-vis absorption spectra of HL-HBPs before and after adsorption-desorption process. The absorption spectra have rarely changed after 10 rounds of adsorption-desorption process, including the characteristic absorption peak of the dyes ($\lambda_{\max, RB} = 553.5$ nm), indicating that dyes are rarely retained on the HL-HBPs. The UV-vis absorption spectra measurements were performed on a microplate reader (Agilent, EPOCH2, USA). HL-HBPs, either original or those obtained after 1st-10th adsorption-desorption process, were dispersed in water for UV-vis absorption spectra measurements from 300 nm to 700 nm.

Supplementary Fig. 13:

Supplementary Fig. 13. ^1H NMR spectra of dyes (RB) before and after adsorption-desorption process. The chemical shift positions have rarely changed after the adsorption-desorption process, indicating that there are rare impurities in the recycled dyes. The NMR measurements were performed on a AscendTM 400 spectrometer (400 MHz, Bruker, Switzerland). Dyes, either before or after adsorption-desorption process, were dissolved in CDCl_3 for ^1H NMR measurement.

Minor/Editing Comments:

1. It is suggested to use the term 'synthetic wastewater' instead of 'mimicked wastewater' throughout this manuscript and figure(s).
2. Line 155: what is ecology pollution? Do you want to say water pollution?
3. Line 157: "To treat the vast organic dye wastewater" – revise this sentence and make it simple.

4. Line 162-164: remove the word 'mediated'
5. Line 168: Change 'Common-use' to 'Commonly-used'
6. Line 232: change 'second' to 's'
7. suggestion to change 'heterostructured' to 'heterostructure'

Response: We have revised the manuscript according to the reviewer's kind suggestions.

1. The term 'mimicked wastewater' has been revised to 'synthetic wastewater' throughout the manuscript and figures (Page 4, Line 79; Page 22, Line 420; Fig. 6a; Supplementary Fig. 16).
2. Line 193 in revised manuscript: Yes, we want to say water pollution. 'ecology pollution' has been revised to 'water pollution'.
3. Line 194 in revised manuscript: "To treat the vast organic dye wastewater" has been revised to 'To treat dyed wastewater'.
4. Line 202-203 in revised manuscript: The word 'mediated' has been removed.
5. Line 209 in revised manuscript: 'common-use' has been revised to 'commonly-used'.
6. Line 279 in revised manuscript: 'second' has been revised to 's'.
7. The term 'heterostructured' has been revised to 'heterostructure' throughout the manuscript, figures, and supplementary information (For example, Line 1 and Line 22 in revised manuscript).

Response to Reviewer #2:

Reviewer #2 (Remarks to the Author):

In this study, the unprecedented omnidispersity of hydrophilic-hydrophobic heterostructured particles (HL-HBPs) was synthesized. A surface heterogeneous nanostructuring strategy was used. Adhesion force images and Photo-induced force microscopy (PiFM) indicate the heterogeneous distributions on the surface. Organic dyes can be adsorbed from water and release them into organic solvents. Several questions need to be clarified.

Response: We greatly appreciate the reviewer's positive comments.

1. More discussions about the advantages and disadvantages of surface heterogeneous nanostructuring strategy should be added.

Response: Thanks for the reviewer's suggestion. We have added the advantages and disadvantages of surface heterogeneous nanostructuring strategy in the revised manuscript.

Page 10, Line 182-188 in revised manuscript:

“The surface heterogeneous nanostructuring strategy demonstrates superior advantages. First, it avoids the import of foreigner organic surfactants or nanostructures, especially those obtained after a complicated process. Second, it achieves omnidispersion in various solvents with wide-range polarity, which is only realized with rare examples of existing materials. Even with these advantages, this strategy is only proved by the emulsion interfacial polymerization in functional particle synthesis. Its generality and uniqueness remain to be explored in other functional material systems.”

2. The novelty of this study should be added in the abstract.

Response: Thanks for the reviewer's suggestion. We have added the novelty of this study in the abstract.

Page 2, Line 29-34 in revised manuscript:

“The surface heterogeneous nanostructuring strategy provides an unconventional approach to achieve omnidispersion of colloidal particles beyond surface modification, and the omnidispersible HL-HBPs demonstrate superior capability for dye recycle merely by solvent exchange. These omnidispersible HL-HBPs show great potentials in industrial process and environmental protection.”

3. More background of separation the organic dyes should be added.

Response: Thanks for the reviewer's suggestion. We have revised and added more background of the separation of organic dyes in the revised manuscript. The revised

background is described as follows.

Page 11, Line 194-206 in revised manuscript:

“To treat dyed wastewater, most existing strategies are mainly concentrated on the removal of dyes from water. Conventional coagulation-flocculation method can remove part of the organic dyes from wastewater, after which the dye-containing sludge is always dumped^{45, 46}. In addition, this approach is mainly suitable for water insoluble dyes, but generally not suitable for water soluble dyes. Alternatively, biological degradation uses biological materials, such as algae, bacteria, fungi, and yeasts, to disintegrate organic dyes. This approach also generates biomass sludge⁴⁷. State-of-the-art methods seek to remove organic dyes from water maximumly and more sustainably, such as catalytic oxidation⁴⁸, membrane nanofiltration^{49, 50}, and porous particle adsorption^{51, 52}. Organic dyes are degraded into smaller molecules, excluded from water, or extracted from water. These catalysts, membranes, and adsorbent materials can be repeatedly used after desorption or regeneration.”

4. Page 5 line 99 and (and 1,451 cm^{-1}), and 1,735 cm^{-1} should be checked.

Response: Thanks for the reviewer’s kind reminding. We checked the wavenumbers again by retrieving more references, and confirmed that 1,451 cm^{-1} and 1,735 cm^{-1} can be attributed to characteristic peaks of PS and epoxy resin, respectively.

Figure R1. FTIR spectra of PS and PS nanocomposites (Figure 2 in literature) (*Physica B* **533**, 12-16 (2018)). In the literature, the absorption peak of 1451 cm^{-1} was attributed to C=C group of benzene ring on PS. We cited the reference as ref. 32 in the revised manuscript.

References:

32. Bhavsar, S., Patel, G.B. & Singh, N.L. Investigation of optical properties of aluminium oxide doped polystyrene polymer nanocomposite films. *Physica B* **533**, 12-16 (2018).

Figure R2. a, FTIR spectra (Figure 1 in literature), **b**, FTIR spectral assignments (Table 1 in literature) of epoxy resin (*J. Polym. Sci., Part A: Polym. Chem.* **38**, 2934-2944 (2000)). In the literature, the absorption peak of 1731 cm^{-1} was attributed to ester group of epoxy resin, and the absorption peak of 1751 cm^{-1} (or $1,753 \text{ cm}^{-1}$) was attributed to $\text{C}=\text{O}$ group. Therefore, it is reasonable that the absorption peak of $1,735 \text{ cm}^{-1}$ in our study can be attributed to ester group of epoxy resin. We cited the reference as ref. 33 in the revised manuscript.

References:

33. Lin, R.-H. In situ FTIR and DSC investigation on cure reaction of liquid aromatic dicyanate ester with different types of epoxy resin. *J. Polym. Sci., Part A: Polym. Chem.* **38**, 2934-2944 (2000).

5. Procedures for Photo-induced force infrared spectra need to be added.

Response: Thanks for the reviewer's kind suggestion. We added the procedures for photo-induced force infrared spectra in the part of Methods-PiFM infrared spectra and imaging. The process was performed and described by one of the authors, Padraic

O'Reilly, a PiFM expert from Molecular Vista Inc. (CA, USA).

Methods (Page 24, Line 478-Page 25, Line 482, in revised manuscript):

“PiFM spectra were taken with a pitch of 16 nm, an acquisition time of 20 s and were power normalized. For spectral acquisition, the laser sweeps through its full range with a dwell time of ~ 17 ms/cm⁻¹ (spectral range/time per spectrum), during which the probe records the PiFM response at the region of interest with sub ~ 10 nm resolution.”

6. Some abbreviations should be listed in tables, such as PSS, PSDVB, PiFM and so on.

Response: Thanks for the reviewer’s good suggestion. We have added a table in the Supplementary Information of revised manuscript listing important abbreviations and corresponding full names, as shown in Supplementary Table 1.

Supplementary Table 1:

Methods (Page 22, Line 430-431, in revised manuscript):

“**Abbreviations.** Important abbreviations and corresponding full names were listed in Supplementary Table 1.”

Supplementary Table 1. Important abbreviations and corresponding full names.

Abbreviation	Full name
HL-HBPs	Hydrophilic-hydrophobic heterostructure particles
HLPs	Hydrophilic particles
HBPs	Hydrophobic particles
PSS	Poly(sodium 4-styrenesulfonate)
PSDVB	Poly(styrene-divinyl benzene)
PTMAEMC	Poly(2-trimethylammoniummethyl methacrylate chloride)
PiFM	Photo-induced force microscopy
SEM	Scanning electron microscopy
TEM	Transmission electron microscopy

AFM	Atomic force microscopy
LSCM	Laser scanning confocal microscopy
NMR	Nuclear magnetic resonance
BET	Brunauer-Emmett-Teller
MOFs	Metal organic frameworks
LDHs	Layered double hydroxides
DLVO	Derjaguin-Landau-Verwey-Overbeek
ζ	Zeta potential
η	Adsorption depth
κ^{-1}	Debye length

7. Page 17 line 330 The dye adsorption efficiency slowly decreases in the several initial rounds, reasons should be given.

Response: Thanks for the reviewer's helpful comments and valuable suggestions. It is speculated that the decrement of dye adsorption efficiency in initial rounds can be attributed to the retainment of minority of the dyes on HL-HBPs, probably caused by the existence of dead pores in HL-HBPs. To reduce the retainment of organic dyes and further enhance the recycle efficiency, we added a post-treatment process for HL-HBPs to improve their pore interconnectivity by eliminating dead pores. In a typical post-treatment process, HL-HBPs were dispersed in DCM to dissolve and remove the linear PS, which was introduced as template particles during the synthesis process of HL-HBPs. Resultantly, the dye adsorption efficiency and desorption efficiency maintain 93~96% and 94~98%, respectively, in 10 dye recycle rounds (Fig. 6c). We greatly appreciate that the reviewer has proposed the concern, which helps us to recognize the important issue of recycle efficiency. We believe the quality of our work has improved significantly after a careful revision according to the reviewer's valuable suggestions. Fig. 6c has been replaced with improved dye recycle performance and post-treatment

process for HL-HBPs has been added in revised manuscript.

Page 20, Line 396-Page 21, Line 397, in revised manuscript:

“In 10 dye recycle rounds, the dye adsorption efficiency and desorption efficiency maintain 93~96% and 94~98%, respectively (Fig. 6c).”

Fig. 6c

Fig. 6 | Omnidispersity-dependent solvent exchange assisted dye recycle using the HL-HBPs. c, Dye adsorption efficiency in water and desorption efficiency in organic solvents using HL-HBPs at different recycle rounds. Dye adsorption and desorption efficiency, Mean \pm SD, n = 3.

Methods (Page 23, Line 460-Page 24, Line 465, in revised manuscript):

“*Post-treatment of HL-HBPs.* The pore interconnectivity of the synthesized HL-HBPs can be additionally improved by dissolving and removing the linear PS, which was introduced as template particles during the synthesis process of HL-HBPs. Typically, HL-HBPs were dispersed in DCM for 1 h, and then isolated from DCM by centrifugation. This process was repeated for three times. Finally, the resultant particles were thoroughly dried at ambient temperature.”

8. Page 13 line 252 The kinetic analysis (figures) for the removal of dye in the adsorption process should be given. The comparison of similar materials or conventional materials need to be added.

Response: Thanks for the reviewer’s suggestion. We tested the dye adsorption performance at different concentration of HL-HBPs (from 1.33 mg mL⁻¹ to 3.33 mg

mL⁻¹, 6.67 mg mL⁻¹, 10 mg mL⁻¹, and 13.33 mg mL⁻¹) to analyze the adsorption kinetics. The adsorption equilibrium was almost achieved at adsorption time of 5 s, and adsorption efficiency rarely increased in 5 minutes (Fig. 5b in revised manuscript). In addition, we have added Supplementary Table 4 in the Supplementary Information of revised manuscript to compare the dye desorption performance of HL-HPBs with other materials.

Page 15, Line 279-283 in revised manuscript:

“Quantificationally, the dye adsorption efficiency was tested for kinetics study when the concentration of HL-HPBs varied from 1.33 mg mL⁻¹ to 3.33 mg mL⁻¹, 6.67 mg mL⁻¹, 10 mg mL⁻¹, and 13.33 mg mL⁻¹. The adsorption equilibrium was almost achieved at adsorption time of 5 s, and adsorption efficiency rarely increased in 5 min (Fig. 5b).”

Fig. 5b:

Fig. 5 | Dye adsorption kinetics and mechanism with the HL-HPBs. b, RB adsorption efficiency with PSS-PSDVB HL-HPBs of various particle concentration (C_{HL-HPBs}) at different adsorption time. The initial concentration of RB solution is 20 ppm. Dye adsorption efficiency, Mean ± SD, n = 3.

Supplementary Table 4:

Supplementary Table 4. Comparison of the pore size, BET surface area, dye adsorption capacity, and dye adsorption equilibrium time/kinetics of HL-HPBs with materials reported in literatures.

Materials	Pore size	BET surface area	Adsorption capacity	Adsorption equilibrium time/kinetics	Ref.
MOFs ([In ₃ O(COO) ₆] ^{+/-} -based)	0.28 nm~1.37 nm	1078 m ² g ⁻¹	20:1 in molar ratio (MOF:dye)	16 h~64 h	7
MOFs (NH ₂ -UiO-66)	~1 nm	1035 m ² g ⁻¹	Up to 697.7 mg g ⁻¹	Langmuir K _L : 0.006 g mg ⁻¹ min ⁻¹	8
Activated carbons (from woods)	/	/	~10 mg g ⁻¹	45 min	9
Activated carbons (Calgon, USA)	1.078 nm~1.088 nm	972 m ² g ⁻¹ 1~1015 m ² g ⁻¹	Up to 1.4 mmol g ⁻¹	> 700 h	10
Ferromagnetic hierarchical porous carbon	Wide range	260 m ² g ⁻¹	0.16 mg m ⁻²	~2 h	11
Carbon particles	1~10 nm	5.2 m ² g ⁻¹	Up to 79.5 mg g ⁻¹	10 s	12
Covalent organic polymers	0.51 nm, 0.76 nm, 1.36 nm	479 m ² g ⁻¹	/	30 min	13
LDHs (MgAl-LDH)	/	/	Up to 186 mg g ⁻¹	5 min	14
LDHs (NiAl-LDH)	3.4 nm	97 m ² g ⁻¹	150 mg g ⁻¹	6 min	15
Carbon-doped boron nitride	/	18.7 m ² g ⁻¹	Up to 747.10 mg g ⁻¹	2 h~3 h	16
MgO	/	154.85 m ² g ⁻¹	Up to 549.45 mg g ⁻¹	> 20 min	17
Chitosan hydrogel	4.34 nm~7.10 nm	2.15 m ² g ⁻¹ 1~42.67 m ² g ⁻¹	Up to 1836 mg g ⁻¹	> 300 min	18

	nm	1				
Amino grafted MCM-41	~3.4 nm	/		Up to 300 mg g ⁻¹	> 5 min	19
HL-HBPs	1 nm~100 nm	7.6 m ² g ⁻¹ ~14.8 m ² g ⁻¹		Up to 13.62 mg g ⁻¹	5 s	This work

References in Supplementary Information:

- Zhao, X. et al. Selective anion exchange with nanogated isoreticular positive metal-organic frameworks. *Nat. Commun.* **4**, 2344 (2013).
- Wang, H. et al. Membrane adsorbers with ultrahigh metal-organic framework loading for high flux separations. *Nat. Commun.* **10**, 4204 (2019).
- Heibati, B. et al. Kinetics and thermodynamics of enhanced adsorption of the dye AR 18 using activated carbons prepared from walnut and poplar woods. *J. Mol. Liq.* **208**, 99-105 (2015).
- Wang, S. & Zhu, Z. Effects of acidic treatment of activated carbons on dye adsorption. *Dyes Pigments* **75**, 306-314 (2007).
- Wang, D.-W., Li, F., Lu, G.Q. & Cheng, H.-M. Synthesis and dye separation performance of ferromagnetic hierarchical porous carbon. *Carbon* **46**, 1593-1599 (2008).
- Seifikar, F., Azizian, S. & Sillanpaa, M. Microwave-assisted synthesis of carbon powder for rapid dye removal. *Mater. Chem. Phys.* **250**, 123057 (2020).
- Byun, J., Patel, H.A., Thirion, D. & Yavuz, C.T. Charge-specific size-dependent separation of water-soluble organic molecules by fluorinated nanoporous networks. *Nat. Commun.* **7**, 13377 (2016).
- Sansuk, S., Srijaranai, S. & Srijaranai, S. A new approach for removing anionic organic dyes from wastewater based on electrostatically driven assembly. *Environ. Sci. Technol.* **50**, 6477-6484 (2016).
- Pahalagedara, M.N. et al. Removal of azo dyes: Intercalation into sonochemically synthesized NiAl layered double hydroxide. *J. Phys. Chem. C* **118**, 17801-17809 (2014).
- Wang, P., Wang, P., Guo, Y., Rao, L. & Yan, C. Selective recovery of protonated dyes from dye wastewater by pH-responsive BCN material. *Chem. Eng. J.* **412**,

128532 (2021).

17. Cao, N. et al. Superior selective adsorption of MgO with abundant oxygen vacancies to removal and recycle reactive dyes. *Sep. Purif. Technol.* **275** (2021).
18. Liu, Y. et al. Efficient removal and recycle of acid blue 93 dye from aqueous solution by acrolein crosslinked chitosan hydrogel. *Colloids Surf. Physicochem. Eng. Aspects* **632** (2022).
19. Rizzi, V. et al. Amino grafted MCM-41 as highly efficient and reversible ecofriendly adsorbent material for the Direct Blue removal from wastewater. *J. Mol. Liq.* **273**, 435-446 (2019).

9. The adsorption mechanism should be added.

Response: Thanks for the reviewer's kind suggestion. The dye adsorption can be attributed to the synergy of electrostatic interaction, hydrogen bonding interaction, hydrophobic/ π - π bonding interaction. Detailed adsorption mechanism has been added in the revised manuscript, and the role of hydrophilic domains has been further clarified in Supplementary Fig. 11.

Page 18, Line 343-353 in revised manuscript:

“The outer hydrophilic domains are composed of negatively charged PSS with thickness of tens of nanometers, which provide intense electrostatic field for strong electrostatic interaction (Supplementary Fig. 11). Positively charged organic dyes in a long range ($\sim 1 \mu\text{m}$) can be attracted immediately to neutralize the charge of hydrophilic domains. Upon attracted towards the HL-HBPs, the hydrophobic domains on HL-HBPs provide short-range hydrophobic/ π - π bonding interaction. Most of the pores exhibit pore diameter from 10 nanometers to 100 nanometers (Supplementary Fig. 4c,d), which allows rapid diffusion and adsorption of organic dye molecules. When the surface charge of HL-HBPs is neutralized, the diffusion of organic dyes towards the particles is almost halted, and adsorption equilibrium is achieved.”

Supplementary Fig. 11

Supplementary Fig. 11. Comparison of dye adsorption kinetics for HL-HBPs prepared with different feed amount of negatively charged hydrophilic monomer.

a, Dye adsorption efficiency of HL-HBPs with varied SS feed amount at different time. Mean \pm SD, $n = 3$.

b, Dye adsorption efficiency of HL-HBPs with varied SS feed amount at adsorption time of 5 s. Mean \pm SD, $n = 3$.

The feed amount of SS (50, 100, 250, 500, and 750 mg) was varied in the emulsion interfacial polymerization process for the synthesis of HL-HBPs. Resultant HL-HBPs were compared for dye adsorption efficiency at different adsorption time. For all HL-HBPs, the dye adsorption efficiency was almost achieved in 5 s. As the SS feed amount increases from 50 mg to 250 mg, dye adsorption efficiency of corresponding HL-HBPs increases at adsorption time of 5 s. Further increasing the feed amount of negatively charged hydrophilic monomer (SS) to 500 mg and 750 mg has little influence on the dye adsorption efficiency. These results indicate that the dye adsorption is highly dependent on the negatively charged hydrophilic domain (PSS) content. These hydrophilic domains could provide intense electrostatic field for strong electrostatic interaction, facilitating rapid dye adsorption.

Response to Reviewer #3:

Reviewer #3 (Remarks to the Author):

In this work, the authors present a general strategy for heterogeneous surface nanostructuring for the preparation of particles with omnidispersibility in various

solvents that can be used for recycling organic dyes. The strategy is based on the rational design of particles containing both hydrophilic and hydrophobic segments resulting from the heterogeneous emulsion polymerization process. As a result, the obtained HL-HPBs exhibit unique dispersion behavior in organic solvents and aqueous solutions, which is crucial for the rapid adsorption and desorption of organic dyes and the regeneration of the particles. Overall, the results of this work are well presented from both fundamental and technological points of view. The paper is well written. I would recommend the publication of this work in Nature Communications if the following points are addressed:

Response: We greatly appreciate the reviewer's positive comments.

1. The authors need to discuss in more detail why the adsorption/desorption efficiency of dyes decreases with increasing number of cycles. 84% adsorption efficiency after the 10th round doesn't seem that great, how can the recycling efficiency be further improved by the chemical design of the particles?

Response: Thanks for the reviewer's helpful comments and valuable suggestions. It is speculated that the decrement of dye adsorption efficiency in initial rounds can be attributed to the retainment of minority of the dyes on HL-HBPs, probably caused by the existence of dead pores in HL-HBPs. To reduce the retainment of organic dyes and further enhance the recycle efficiency, we added a post-treatment process for HL-HBPs to improve their pore interconnectivity by eliminating dead pores. In a typical post-treatment process, HL-HBPs were dispersed in DCM to dissolve and remove the linear PS, which was introduced as template particles during the synthesis process of HL-HBPs. Resultantly, the dye adsorption efficiency and desorption efficiency maintain 93~96% and 94~98%, respectively, in 10 dye recycle rounds (Fig. 6c). We greatly appreciate that the reviewer has proposed the concern, which allows us to recognize the important issue of recycle efficiency. We believe the quality of our work has improved significantly after a careful revision according to the reviewer's valuable suggestions. Fig. 6c has been replaced with improved dye recycle performance and post-treatment

process for HL-HBPs has been added in revised manuscript.

Page 20, Line 396-Page 21, Line 397, in revised manuscript:

“In 10 dye recycle rounds, the dye adsorption efficiency and desorption efficiency maintain 93~96% and 94~98%, respectively (Fig. 6c).”

Fig. 6c

Fig. 6 | Omnidispersity-dependent solvent exchange assisted dye recycle using the HL-HBPs. c, Dye adsorption efficiency in water and desorption efficiency in organic solvents using HL-HBPs at different recycle rounds. Dye adsorption and desorption efficiency, Mean \pm SD, n = 3.

Methods (Page 23, Line 460-Page 24, Line 465, in revised manuscript):

“*Post-treatment of HL-HBPs.* The pore interconnectivity of the synthesized HL-HBPs can be additionally improved by dissolving and removing the linear PS, which was introduced as template particles during the synthesis process of HL-HBPs. Typically, HL-HBPs were dispersed in DCM for 1 h, and then isolated from DCM by centrifugation. This process was repeated for three times. Finally, the resultant particles were thoroughly dried at ambient temperature.”

2. The authors should provide a general table to compare the adsorption/desorption performance of dyes using HL-HPBs with other state-of-art materials in the literature in SI.

Response: Thanks for the reviewer’s suggestion. We have added two tables, Supplementary Table 4 and Supplementary Table 6, in the Supplementary Information

of revised manuscript. Supplementary Table 4 compares the dye adsorption performance, and Supplementary Table 6 compares the dye desorption performance of HL-HPBs with other state-of-art materials, respectively.

Supplementary Table 4:

Supplementary Table 4. Comparison of the pore size, BET surface area, dye adsorption capacity, and dye adsorption equilibrium time/kinetics of HL-HPBs with materials reported in literatures.

Materials	Pore size	BET surface area	Adsorption capacity	Adsorption equilibrium time/kinetics	Ref.
MOFs ([In ₃ O(COO) ₆] ⁺ -based)	0.28 nm~1.37 nm	1078 m ² g ⁻¹	20:1 in molar ratio (MOF:dye)	16 h~64 h	7
MOFs (NH ₂ -UiO-66)	~1 nm	1035 m ² g ⁻¹	Up to 697.7 mg g ⁻¹	Langmuir K _L : 0.006 g mg ⁻¹ min ⁻¹	8
Activated carbons (from woods)	/	/	~10 mg g ⁻¹	45 min	9
Activated carbons (Calgon, USA)	1.078 nm~1.088 nm	972 m ² g ⁻¹ ~1015 m ² g ⁻¹	Up to 1.4 mmol g ⁻¹	> 700 h	10
Ferromagnetic hierarchical porous carbon	Wide range	260 m ² g ⁻¹	0.16 mg m ⁻²	~2 h	11
Carbon particles	1~10 nm	5.2 m ² g ⁻¹	Up to 79.5 mg g ⁻¹	10 s	12
Covalent organic polymers	0.51 nm, 0.76 nm, 1.36 nm	479 m ² g ⁻¹	/	30 min	13

LDHs (MgAl-LDH)	/	/	Up to 186 mg g ⁻¹	5 min	14
LDHs (NiAl-LDH)	3.4 nm	97 m ² g ⁻¹	150 mg g ⁻¹	6 min	15
Carbon-doped boron nitride	/	18.7 m ² g ⁻¹	Up to 747.10 mg g ⁻¹	2 h~3 h	16
MgO	/	154.85 m ² g ⁻¹	Up to 549.45 mg g ⁻¹	> 20 min	17
Chitosan hydrogel	4.34 nm~7.10 nm	2.15 m ² g ⁻¹ ~42.67 m ² g ⁻¹	Up to 1836 mg g ⁻¹	> 300 min	18
Amino grafted MCM-41	~3.4 nm	/	Up to 300 mg g ⁻¹	> 5 min	19
HL-HBPs	1 nm~100 nm	7.6 m ² g ⁻¹ ~14.8 m ² g ⁻¹	Up to 13.62 mg g ⁻¹	5 s	This work

Supplementary Table 6:

Supplementary Table 6. Comparison of the dye recycle performance for HL-HBPs and materials in literatures. To desorb organic dyes from materials reported in literatures, eluents containing inorganic acid, alkaline, or salt, are often added into the aqueous solution to weaken the interactions between materials and dyes. Such eluents make the dye recycle more complicated. In comparison, organic solvent is used for dye desorption, and dyes can be recycled under a simple distillation process.

Materials	Recyclability of materials	Recyclability of dyes	Eluents	Ref.
MOFs	Yes	Not available	Salt (NaNO ₃)	7, 8
LDHs	Yes	Not available	Salt containing Cl ⁻ , NO ₃ ⁻ , or CO ₃ ²⁻	14, 15

Carbon-doped boron nitride	Yes	Not available	Acid (HCl)	16
MgO	Yes	Yes	Acid (HCl)	17
Chitosan hydrogel	Yes	Not available	Alkali (NaOH)	18
Amino grafted MCM-41	Yes	Not available	Alkali (NaOH)	19
HL-HBPs	Yes	Yes	Organic solvent	This work

References in Supplementary Information:

- Zhao, X. et al. Selective anion exchange with nanogated isoreticular positive metal-organic frameworks. *Nat. Commun.* **4**, 2344 (2013).
- Wang, H. et al. Membrane adsorbers with ultrahigh metal-organic framework loading for high flux separations. *Nat. Commun.* **10**, 4204 (2019).
- Heibati, B. et al. Kinetics and thermodynamics of enhanced adsorption of the dye AR 18 using activated carbons prepared from walnut and poplar woods. *J. Mol. Liq.* **208**, 99-105 (2015).
- Wang, S. & Zhu, Z. Effects of acidic treatment of activated carbons on dye adsorption. *Dyes Pigments* **75**, 306-314 (2007).
- Wang, D.-W., Li, F., Lu, G.Q. & Cheng, H.-M. Synthesis and dye separation performance of ferromagnetic hierarchical porous carbon. *Carbon* **46**, 1593-1599 (2008).
- Seifikar, F., Azizian, S. & Sillanpaa, M. Microwave-assisted synthesis of carbon powder for rapid dye removal. *Mater. Chem. Phys.* **250**, 123057 (2020).
- Byun, J., Patel, H.A., Thirion, D. & Yavuz, C.T. Charge-specific size-dependent separation of water-soluble organic molecules by fluorinated nanoporous networks. *Nat. Commun.* **7**, 13377 (2016).
- Sansuk, S., Srijaranai, S. & Srijaranai, S. A new approach for removing anionic organic dyes from wastewater based on electrostatically driven assembly. *Environ. Sci. Technol.* **50**, 6477-6484 (2016).
- Pahalagedara, M.N. et al. Removal of azo dyes: Intercalation into

- sonochemically synthesized NiAl layered double hydroxide. *J. Phys. Chem. C* **118**, 17801-17809 (2014).
16. Wang, P., Wang, P., Guo, Y., Rao, L. & Yan, C. Selective recovery of protonated dyes from dye wastewater by pH-responsive BCN material. *Chem. Eng. J.* **412**, 128532 (2021).
 17. Cao, N. et al. Superior selective adsorption of MgO with abundant oxygen vacancies to removal and recycle reactive dyes. *Sep. Purif. Technol.* **275** (2021).
 18. Liu, Y. et al. Efficient removal and recycle of acid blue 93 dye from aqueous solution by acrolein crosslinked chitosan hydrogel. *Colloids Surf. Physicochem. Eng. Aspects* **632** (2022).
 19. Rizzi, V. et al. Amino grafted MCM-41 as highly efficient and reversible ecofriendly adsorbent material for the Direct Blue removal from wastewater. *J. Mol. Liq.* **273**, 435-446 (2019).

3. For the adsorption index, it is not entirely clear if the interaction sites (electrostatic, hydrogen bonding, hydrophobic) contribute equally to this calculation? Basically, I would expect them to have different binding abilities with the charged/functional groups in the particles. Further discussion would be needed to distinguish the role of the different interaction sites.

Response: Thanks for the reviewer's suggestion. During the calculation process in original manuscript, we assumed that different types of interactions contribute equally. As the reviewer commented, they probably have different binding abilities and roles with the charged/functional groups in the particles. In a long range ($\sim 1 \mu\text{m}$), organic dyes can be attracted by the HL-HBPs *via* electrostatic interaction. Upon attracted towards the HL-HBPs, organic dyes can be adsorbed by the hydrophilic domains *via* electrostatic interaction/hydrogen bonding interaction, and by the hydrophobic domains *via* hydrophobic/ π - π bonding interaction. The interaction energy of electrostatic interaction ($\sim 100 \text{ kT}$) is much higher than that of other interactions, including hydrogen bonding interaction ($5\sim 10 \text{ kT}$) and hydrophobic/ π - π bonding interaction ($\sim 1 \text{ kT}$) (Israelachvili, J.N. Intermolecular and surface forces. (Academic Press, 2011)). Nevertheless, it remains a difficult task to clearly and quantitatively define the

contribution factor of different types of interactions for dye adsorption due to the unique heterostructure. In our manuscript, the adsorption index is an immature attempt to explain the difference of dye adsorption amounts. We expect this index can be further modified for more accurate explanation and prediction as the development of experimental techniques and fundamental theories.

Page 14, Line 262-270 in revised manuscript:

“It should also be noted that the interaction energy of electrostatic interaction (~ 100 kT) is much higher than that of other interactions, including hydrogen bonding interaction ($5\sim 10$ kT) and hydrophobic/ π - π bonding interaction (~ 1 kT)⁸. Nevertheless, it remains a difficult task to clearly and quantitatively define the contribution factor of different types of interactions for dye adsorption due to the unique heterostructure. The adsorption index is an immature attempt to explain the difference of dye adsorption amounts. Further modification is expected for more accurate explanation and prediction as the development of experimental techniques and fundamental theories.”

Reviewers' Comments:

Reviewer #1:

Remarks to the Author:

This is a revised version of the manuscript previously reviewed by this reviewer. Although the authors have made significant efforts to revise the manuscript, this reviewer finds that the underlying reasons for the original recommendation (i.e., reject) still hold. The decision that this reviewer has to make is that is the heterogeneous nanostructuring strategy to obtain omnidispersity novel or an extension (i.e., incremental) of authors' previous work? This reviewer believes that the novelty and application of this incremental work do not rise to the level warranting publication in Nature Communication. In addition, the claims made in the study about the significance of using organic solvents instead of inorganic solvent during dye recycling is weak and does not provide solid ground for the publication of this article. Finally, as pointed out in previous review comments, the material adsorption capacity is very poor and would be a major hurdle in their practical/industrial application. There are many issues that have not been adequately addressed. Some of them are outlined below:

Is a BET surface area of less than 15 m²/g considered good? It remains a challenge to estimate the pore size of materials with a surface area of less than 50 m²/g. This is the reason why it is important to explain why a certain model is used for pore size estimation.

The contact angle data clearly shows that HL-HBPs are clearly hydrophobic, which raises the question about the electrostatic interaction (dye removal mechanism claimed by the author). There is a need to discuss the implications of the achieved results.

The authors claimed that the adsorption rate is rapid. The standard is to compare adsorption performance based on adsorption capacity data (mg/g) rather than adsorption rate. This is because experimental conditions (adsorbent concentration, dye concentration, stirring speed, contact time) affect the adsorption rate. Hence, the reasoning provided by the authors is weak.

"According to our results, the recovered dyes have rarely changed after the adsorption-desorption process, implying their successful recycling. Therefore, our research provides a green approach to achieve repeated use of organic dyes, which is beneficial for reducing organic dye pollution in the printing and dyeing industry." – Mechanisms remain still unclear, especially how to disperse a hydrophobic material (contact angle around 120 degrees).

Without going back and forth (as it is not scientific brainstorming), this reviewer does not recommend the publication of this work.

Reviewer #2:

Remarks to the Author:

The authors have revised based on the suggestions.

Reviewer #3:

Remarks to the Author:

In the revised manuscript, the authors have put additional efforts to address my major concerns. The overall recycling performance has been largely improved. On this basis, I would recommend the publication of this work as it is.

Point-by-point Response

We greatly appreciate the reviewers' important comments and helpful suggestions that significantly improved the quality of our revised manuscript.

Response to Reviewer #1:

Reviewer #1 (Remarks to the Author):

This is a revised version of the manuscript previously reviewed by this reviewer. Although the authors have made significant efforts to revise the manuscript, this reviewer finds that the underlying reasons for the original recommendation (i.e., reject) still hold. The decision that this reviewer has to make is that is the heterogeneous nanostructuring strategy to obtain omnidispersity novel or an extension (i.e., incremental) of authors' previous work? This reviewer believes that the novelty and application of this incremental work do not rise to the level warranting publication in Nature Communication. In addition, the claims made in the study about the significance of using organic solvents instead of inorganic solvent during dye recycling is weak and does not provide solid ground for the publication of this article. Finally, as pointed out in previous review comments, the material adsorption capacity is very poor and would be a major hurdle in their practical/industrial application. There are many issues that have not been adequately addressed. Some of them are outlined below.

Response: We greatly appreciate the reviewer's important comments that further deepen our understanding on our study. We would like to response the main concerns raised by the reviewer.

Firstly, we believe that the present work has significant novelty compared with our previous works. It is a great challenge to prepare omnidispersible colloidal particles, although several examples have been demonstrated by introduction of nanoprotusions. In present work, we have proposed a surface heterogeneous nanostructuring strategy to realize the omnidispersion of particles. This strategy can be realized by the heterostructure particles synthesized by emulsion interfacial polymerization proposed

in our previous works. As the reviewer pointed out, the previous works have mainly concentrated on the separation of trace biomolecules from complex samples. The unique omnidispersity of heterostructure particles enables an unprecedented simultaneous recycle of organic dyes and regeneration of HL-HBPs from the synthetic wastewater merely through solvent exchange. Therefore, we believe this work is not an extension (or increment) of our previous works, and its novelty can meet the high standard and high quality of *Nature Communications*.

Secondly, there is unique significance to use organic solvents instead of inorganic solvent for organic dye recycle. In previous revision, we have systematically summarized and compared the recyclability of adsorbent materials and organic dyes with existing material and our particles (Supplementary Table 6). Organic solvents are easy to be removed from dyes by simple distillation. In contrast, additional steps are required to remove inorganic acid, alkaline, and salt from water. We have added the significance in the revised manuscript.

Page 21, Line 409-412:

“There is unique significance to use organic solvents instead of inorganic solvent for organic dye recycle. Organic solvents are easy to be removed from dyes by simple distillation. In contrast, additional steps are required to remove inorganic acid, alkaline, and salt from aqueous solution of recycled dyes.”

Thirdly, we have underlined the necessity of improving dye adsorption capacity in the revised manuscript. Although the dye adsorption amount of HL-HBPs is lower than those existing materials with high surface area like MOFs, their adsorption rate is rapid (Supplementary Table 4). The next and great challenge is to create adsorbent materials with high adsorption capacity while maintaining the rapid adsorption performance. In our lab, several projects are ongoing from different aspects for this supreme goal.

Major Comments:

1. Is a BET surface area of less than 15 m²/g considered good? It remains a challenge to estimate the pore size of materials with a surface area of less than 50 m²/g. This is

the reason why it is important to explain why a certain model is used for pore size estimation.

Response: We greatly appreciate the reviewer's comments. The pore size is characterized by nitrogen adsorption-desorption test, and then estimated by the Barrett-Joyner-Halenda (BJH) model (Lowell, S., Shields, J. E., Thomas, M. A. & Thommes, M. Characterization of porous solids and powders: surface area, pore size and density. (Springer Science+Business Media New York, 2004)). In this book, we have learned that "Using highly accurate volumetric adsorption equipment, it is possible to measure absolute surface areas as low as approximately 0.5-1 m² with nitrogen as the adsorptive." (Page 79), and that "Among these different approaches, the Barrett-Joyner-Halenda method can be considered as the most popular method for mesopores size analysis." (Page 104). The total surface area of HL-HBPs used for nitrogen adsorption-desorption (2.9-3.9 m²) is higher than that is required (as low as approximately 0.5-1 m²). In addition, the adsorption-desorption isotherms of HL-HBPs indicate a type IV adsorption model, and most of the pores exhibit pore diameter from 10 nanometers to 100 nanometers (Supplementary Fig. 4b, c). Therefore, the pore size estimation results based on BJH model are reliable.

2. The contact angle data clearly shows that HL-HBPs are clearly hydrophobic, which raises the question about the electrostatic interaction (dye removal mechanism claimed by the author). There is a need to discuss the implications of the achieved results.

Response: We appreciate the reviewer's comments and helpful suggestions. We would like to clarify that the contact angle for HL-HBPs adhered tape ($122.6 \pm 1.6^\circ$) does not certainly imply the absence of electrostatic interaction on HL-HBPs. The hydrophilicity-hydrophobicity of the particles was evaluated by a commonly-used approach. Typically, the particles were adhered to a glass microscope slide by double-sided adhesive tape, and three-phase contact angle was measured by the sessile drop method (*Powder Technol.* **233**, 52-64 (2013)). The contact angle can be affected by the coverage of the particles on the tape, the interfacial tension of the particles, and the

surface structure (*Nanoscale* **4**, 2202-2218 (2012); *Appl. Surf. Sci.* **255**, 3371-3374 (2008)). The HL-HBPs adhered tape shows a contact angle of $122.6 \pm 1.6^\circ$, which is much reduced compared with HBPs adhered tape ($136.0 \pm 2.7^\circ$) (Supplementary Fig. 6), suggesting the presence of negatively charged hydrophilic domains (PSS). The ζ values also show the HL-HBPs are highly charged (Supplementary Fig. 2), which guarantee the dispersion of HL-HBPs in water. To illustrate the contact angle results more clearly, we added some discussion after the Figure caption of Supplementary Fig. 6 in the revised manuscript.

Page 11, Figure caption of Supplementary Fig. 6, in revised Supplementary Information:

“We would like to clarify that the contact angle for HL-HBPs adhered tape ($122.6 \pm 1.6^\circ$) does not certainly imply the absence of electrostatic interaction on HL-HBPs. The hydrophilicity-hydrophobicity of the particles was evaluated by a commonly-used approach. Typically, the particles were adhered to a glass microscope slide by double-sided adhesive tape, and three-phase contact angle was measured by the sessile drop method¹. The contact angle can be affected by the coverage of the particles on the tape, the interfacial tension of the particles, and the surface structure^{2, 3}. The HL-HBPs adhered tape shows a contact angle of $122.6 \pm 1.6^\circ$, which is much reduced compared with HBPs adhered tape ($136.0 \pm 2.7^\circ$), suggesting the presence of negatively charged hydrophilic domains (PSS).”

References in Supplementary Information:

1. Nowak, E., Combes, G., Stitt, E.H. & Pacey, A.W. A comparison of contact angle measurement techniques applied to highly porous catalyst supports. *Powder Technol.* **233**, 52-64 (2013).
2. Gao, N. & Yan, Y. Characterisation of surface wettability based on nanoparticles. *Nanoscale* **4**, 2202-2218 (2012).
3. Zhou, X., Guo, X., Ding, W. & Chen, Y. Superhydrophobic or superhydrophilic surfaces regulated by micro-nano structured ZnO powders. *Appl. Surf. Sci.* **255**, 3371-3374 (2008).

3. The authors claimed that the adsorption rate is rapid. The standard is to compare adsorption performance based on adsorption capacity data (mg/g) rather than adsorption rate. This is because experimental conditions (adsorbent concentration, dye concentration, stirring speed, contact time) affect the adsorption rate. Hence, the reasoning provided by the authors is weak.

Response: We greatly appreciate the reviewer’s comments. As the reviewer has commented, experimental conditions, including adsorbent concentration, dye concentration, stirring speed, and contact time, affect the adsorption rate. The experimental conditions are varied in the references. After careful comparison, the dye adsorption efficiency versus contact time is frequently used to characterize the adsorption equilibrium and adsorption rate. In our work, we have adopted this form to evaluate the adsorption equilibrium and adsorption rate. We found that the adsorption equilibrium was almost achieved at adsorption time of 5 s, and adsorption efficiency rarely increased in 5 min (Fig. 5b) when the concentration of HL-HBPs varied from 1.33 mg mL⁻¹ to 3.33 mg mL⁻¹, 6.67 mg mL⁻¹, 10 mg mL⁻¹, and 13.33 mg mL⁻¹. These results indicate that the adsorption equilibrium is independent of particle concentration in this range. We further compared the initial adsorbent dosage and initial dye dosage of HL-HBPs with materials reported in literatures. In the references, the initial adsorbent dosage generally ranges from 0.025 mg mL⁻¹ to 6 mg mL⁻¹, and initial dye dosage generally ranges from 10 ppm to 450 ppm. Our experimental conditions are comparable with them. However, the dye adsorption equilibrium time of our HL-HBPs is much shorter than those reported in the references. Therefore, the conclusion that the adsorption rate is rapid is acceptable.

Supplementary Table 4:

Supplementary Table 4. Comparison of the pore size, BET surface area, dye adsorption capacity, dye adsorption equilibrium time/kinetics, initial adsorbent dosage, and initial dye dosage of HL-HBPs with materials reported in literatures.

Materials	Pore size	BET surface area	Adsorption capacity	Adsorption equilibrium time/kinetics	Initial adsorbent dosage	Initial dye dosage	Ref.
-----------	-----------	------------------	---------------------	--------------------------------------	--------------------------	--------------------	------

MOFs ([In ₃ O(COO) ₆] ⁺ - based)	0.28 nm~1.37 nm	1,078 m ² g ⁻¹	20:1 in molar ratio (MOF:dye)	16 h~64 h	~2.2 mg mL ⁻¹	~35 ppm (~7.9×10 ⁻⁵ M)	10
MOFs (NH ₂ -UiO-66)	~1 nm	1,035 m ² g ⁻¹	Up to 697.7 mg g ⁻¹	Langmuir K _L : 0.006 g mg ⁻¹ min ⁻¹	0.5 mg mL ⁻¹	100 ppm	11
Activated carbons (from woods)	/	/	~10 mg g ⁻¹	45 min	3.2 mg mL ⁻¹	50 ppm	12
Activated carbons (Calgon, USA)	1.078 nm~1.088 nm	972 m ² g ⁻¹ ~1,015 m ² g ⁻¹	Up to 1.4 mmol g ⁻¹	> 700 h	0.025 mg mL ⁻¹	1.2×10 ⁻⁵ M~4.6×10 ⁻⁷ M	13
Ferromagnetic hierarchical porous carbon	Wide range	260 m ² g ⁻¹	0.16 mg m ⁻²	~2 h	1 mg mL ⁻¹	1.5×10 ⁻⁴ M	14
Carbon particles	1~10 nm	5.2 m ² g ⁻¹	Up to 79.5 mg g ⁻¹	10 s	0.25 mg mL ⁻¹	10, 30 ppm	15
Covalent organic polymers	0.51 nm, 0.76 nm, 1.36 nm	479 m ² g ⁻¹	/	30 min	1 mg mL ⁻¹	5×10 ⁻⁵ M	16
LDHs (MgAl-LDH)	/	/	Up to 186 mg g ⁻¹	5 min	/	100 ppm	17
LDHs (NiAl-LDH)	3.4 nm	97 m ² g ⁻¹	150 mg g ⁻¹	6 min	1 mg mL ⁻¹	150 ppm	18
Carbon-doped boron nitride	/	18.7 m ² g ⁻¹	Up to 747.10 mg g ⁻¹	2 h~3 h	0.2 mg mL ⁻¹	150 ppm	19
MgO	/	154.85 m ² g ⁻¹	Up to 549.45 mg g ⁻¹	> 20 min	/	100 ppm	20
Chitosan hydrogel	4.34 nm~7.10 nm	2.15 m ² g ⁻¹ ~42.67 m ² g ⁻¹	Up to 1,836 mg g ⁻¹	> 300 min	/	450 ppm	21

Amino grafted MCM-41	~3.4 nm	/	Up to 300 mg g ⁻¹	> 5 min	0.2~6 mg mL ⁻¹	10, 50 ppm	22
HL-HBPs	1 nm~100 nm	7.6 m ² g ⁻¹ ~14.8 m ² g ⁻¹	Up to 13.62 mg g ⁻¹	5 s	0.67~6.67 mg mL ⁻¹ (total)	10 ppm (2.1×10 ⁻⁵ M, RB) (total)	This work

4. “According to our results, the recovered dyes have rarely changed after the adsorption-desorption process, implying their successful recycling. Therefore, our research provides a green approach to achieve repeated use of organic dyes, which is beneficial for reducing organic dye pollution in the printing and dyeing industry.” – Mechanisms remain still unclear, especially how to disperse a hydrophobic material (contact angle around 120 degrees).

Response: We appreciate the reviewer’s comments. This question is related to the Comment 2. We would like to clarify that the contact angle for HL-HBPs adhered tape ($122.6 \pm 1.6^\circ$) does not certainly imply the absence of electrostatic interaction on HL-HBPs. The hydrophilicity-hydrophobicity of the particles was evaluated by a commonly-used approach. Typically, the particles were adhered to a glass microscope slide by double-sided adhesive tape, and three-phase contact angle was measured by the sessile drop method (*Powder Technol.* **233**, 52-64 (2013)). The contact angle can be affected by the coverage of the particles on the tape, the interfacial tension of the particles, and the surface structure (*Nanoscale* **4**, 2202-2218 (2012); *Appl. Surf. Sci.* **255**, 3371-3374 (2008)). The HL-HBPs adhered tape shows a contact angle of $122.6 \pm 1.6^\circ$, which is much reduced compared with HBPs adhered tape ($136.0 \pm 2.7^\circ$) (Supplementary Fig. 6), suggesting the presence of negatively charged hydrophilic domains (PSS). The ζ values also show the HL-HBPs are highly charged (Supplementary Fig. 2), which guarantee the dispersion of HL-HBPs in water. The DLVO calculations further confirm that the dispersion mechanism (Fig. 3d-f). Additionally, the dispersity is not totally dependent on contact angle, because contact angle shows the collective property of the particles in macroscale, while the dispersion

property is mainly influenced by the interaction potential between two particles in microscale. To illustrate the contact angle results more clearly, we added some discussion after the Figure caption of Supplementary Fig. 6 in the revised manuscript. We expect the discussion can eliminate the concern of “how to disperse a hydrophobic material”. With such discussion, the mechanisms for organic dye adsorption-desorption and recycle is reasonable.

Page 11, Figure caption of Supplementary Fig. 6, in revised Supplementary Information:

“We would like to clarify that the contact angle for HL-HBPs adhered tape ($122.6 \pm 1.6^\circ$) does not certainly imply the absence of electrostatic interaction on HL-HBPs. The hydrophilicity-hydrophobicity of the particles was evaluated by a commonly-used approach. Typically, the particles were adhered to a glass microscope slide by double-sided adhesive tape, and three-phase contact angle was measured by the sessile drop method¹. The contact angle can be affected by the coverage of the particles on the tape, the interfacial tension of the particles, and the surface structure^{2, 3}. The HL-HBPs adhered tape shows a contact angle of $122.6 \pm 1.6^\circ$, which is much reduced compared with HBPs adhered tape ($136.0 \pm 2.7^\circ$), suggesting the presence of negatively charged hydrophilic domains (PSS).”

References in Supplementary Information:

1. Nowak, E., Combes, G., Stitt, E.H. & Pacey, A.W. A comparison of contact angle measurement techniques applied to highly porous catalyst supports. *Powder Technol.* **233**, 52-64 (2013).
2. Gao, N. & Yan, Y. Characterisation of surface wettability based on nanoparticles. *Nanoscale* **4**, 2202-2218 (2012).
3. Zhou, X., Guo, X., Ding, W. & Chen, Y. Superhydrophobic or superhydrophilic surfaces regulated by micro-nano structured ZnO powders. *Appl. Surf. Sci.* **255**, 3371-3374 (2008).

Response to Reviewer #2:

Reviewer #2 (Remarks to the Author):

The authors have revised based on the suggestions.

Response: We greatly appreciate the reviewer's positive comments.

Response to Reviewer #3:

Reviewer #3 (Remarks to the Author):

In the revised manuscript, the authors have put additional efforts to address my major concerns. The overall recycling performance has been largely improved. On this basis, I would recommend the publication of this work as it is.

Response: We greatly appreciate the reviewer's positive comments.

Reviewers' Comments:

Reviewer #1:

Remarks to the Author:

This manuscript presents novel results and demonstrates its effectiveness through rigorous experiments. The authors have done a commendable job in addressing the previous comments and improving the quality of their paper. This time, based on the revision the authors have made and overall progress, I am happy to recommend this manuscript for publication without further review.

Point-by-point Response

We greatly appreciate the reviewers' important comments and helpful suggestions that significantly improved the quality of our revised manuscript.

Response to Reviewer #1:

Reviewer #1 (Remarks to the Author):

This manuscript presents novel results and demonstrates its effectiveness through rigorous experiments. The authors have done a commendable job in addressing the previous comments and improving the quality of their paper. This time, based on the revision the authors have made and overall progress, I am happy to recommend this manuscript for publication without further review.

Response: We greatly appreciate the reviewer's positive comments and previous helpful suggestions that help us to understand our work deeply. According to reviewers' suggestions, the quality of our manuscript has been significantly improved. We believe the revised manuscript reach the high standard and high quality of *Nature Communications*.